# Concrete Examination of 100-Year-Old Bridge Structure above the Kłodnica River Flowing through the Agglomeration of Upper Silesia in Gliwice: A Case Study

**DOI:** 10.3390/ma14040981

**Published:** 2021-02-19

**Authors:** Barbara Słomka-Słupik, Jacek Podwórny, Beata Grynkiewicz-Bylina, Marek Salamak, Bibianna Bartoszek, Wiktoria Drzyzga, Marcel Maksara

**Affiliations:** 1Faculty of Civil Engineering, Silesian University of Technology, 44-100 Gliwice, Poland; Marek.Salamak@polsl.pl (M.S.); wiktdrz087@student.polsl.pl (W.D.); marcmak788@student.polsl.pl (M.M.); 2Łukasiewicz Research Network, Institute of Ceramics and Building Materials, Division of Refractory Materials in Gliwice, 44-100 Gliwice, Poland; j.podworny@icimb.pl; 3Laboratory of Material Engineering and Environment, KOMAG Institute of Mining Technology, 44-101 Gliwice, Poland; bbylina@komag.eu (B.G.-B.); bbartoszek@komag.eu (B.B.)

**Keywords:** bridge durability, heavy metals, Kłodnica river in Gliwice, contamination, concrete destruction, ICP, decalcification

## Abstract

The article analyzes the composition of concrete taken from various elements from a 100-year-old bridge in South Poland, so as to analyze its technical condition. The main methods applied during experimental work were: Designation of pH, free chloride content, salinity, XRD and SEM examinations, as well as metals determination using inductively coupled plasma mass spectrometry (ICP­MS), high-performance liquid chromatography (HPLC)-ICP-MS, and cold-vapor atomic absorption spectroscopy (CV-AAS). The concrete of the bridge was strongly carbonated and decalcified with an extremely high content of chlorides. The pH of the concrete was in a range from 10.5 to 12.0. Acid soluble components were between 9.9% and 17.6%. Typical sulfate corrosion phases of concrete were not detected. Friedels’ salt was found only at the extremity of an arch. The crown block was corroded to the greatest extent. Various heavy metals were absorbed into the concrete, likely from previous centuries, when environmental protection policy was poor. The applied research methodology can be used on bridges exposed to specific external influences. The acquired knowledge can be useful in the management processes of the bridge infrastructure. It can help in making decisions about decommissioning or extending the life cycle of the bridge. This work should also sensitize researchers and decision-makers to the context of “bridge safety”.

## 1. Introduction

Bridge structures are exposed to many factors, which weaken the structure, its elements, concrete, and steel. These factors include environmental (heating, freezing, drying, spraying), mechanical (loads), and chemical (exhaust gases, de-icing salts) influences. Moreover, an individual bridge works in specific conditions depending on its location. This article points to the bridge located over a significantly polluted river due to its location in a well-developed, heavily industrialized region. Therefore, a case will be considered where, apart from general factors that weaken bridge structures, there is in addition of high humidity and the influence of impurities—basic salts that corrode the structure (chlorides and sulphates) and toxic metals in the river water. Contaminants found in river water may be of a natural or of anthropogenic origin, and can cause corrosion to the bridge elements or can simply be absorbed by concrete. The contamination of river water could also have changed over the years. Some of the metals absorbed into the concrete may prove what the river was polluted with, long ago. The issues surrounding the destruction of bridges by water has previously been briefly described [1,2,3,4,5,6,7,8,9], however, to the best of our knowledge, there is no literature concerning the destruction of the bridge’s concrete under the influence of the polluted river water.

### 1.1. Bridge Destruction, Processes, and the Life Cycle of the Bridge

The focus for a post-mortem analysis of bridge destruction is devoted to mechanical features, basically. However, experimental results show that the flexural strength, stiffness, ductility, and energy dissipation capacity are constantly lowered with an increasing degree of corrosion of the reinforcement [1,2]. Corrosion effects have been analyzed for coastal and seawater processes on bridge pierce, beams piers [3,4,5], and piles from a chemical point of view [6]. The thaumasite corrosion of concrete bridge structures was described 20 years ago [7]. However these bridges were not built above rivers, therefore the mechanism of concrete destruction is not similar due to the fact that they did not work in an environment with high constant humidity [7]. Regrettably, a disaster can occur without visible signs of destruction, as Hüthwohl et al. [8] noted in their multi-classification of the detection of bridge defects. Exemplary studies on the recently collapsed Polcevera Bridge in Genoa (Italy) [9] have demonstrated that a hostile environment caused the degradation of the bridge, which developed much faster than expected. Invernizzi et al. [9] highlighted that more attention should be paid to understanding the phenomena that cause very-high cycle corrosion fatigue in existing civil infrastructures.

The chemical destruction of the bridge structural elements is caused mainly by chloride, sulfate, and hydrocarbon ions. Chloride ions that corrode reinforcing steel are present in anti-frost salts and in river water. Sulphate ions, in turn, can originate from groundwater and river water, and can cause concrete cracks and softening. Hydrocarbon ions lower pH and form carbonates. They come from pumped mine underground (brine) water into the river water. Indicators demonstrating anthropogenic pollution are found also in river bottom sediments. Of course, some of them could have been removed or displaced along with the river current. Therefore the authors assume that the presence of certain impurities from previous years can be elucidated by testing the concrete, as these impurities were absorbed into the concrete matrix. Confirming this assumption is the premise of the article. However, this is a supporting study. The accumulation of toxic metals in concrete is possible due to thermal transport processes of solutions in porous materials, including concrete. This transport takes place as a result of capillary suction, ion diffusion, and water evaporation from concrete. Salts accumulate on surfaces forming efflorescence, weeping, exfoliation, stalactites, and crusts [10,11,12,13,14,15,16,17,18,19,20,21,22,23,24,25].

The life cycle of the bridge is the time interval that begins with the first concept of passing through an obstacle (for example river, road) and includes its design, construction, and use, until its demolition. It contains four primary phases: 1. Preparatory (concept, design); 2. investment (construction, commissioning); 3. operational (use, maintenance, repair, rehabilitation), and 4. decommissioning (demolition and utilization) [26]. The life cycle of a bridge is increasingly associated with BIM (Building Information Management) [27] and IAM (Infrastructure Asset Management) [28]. For the bridge in Gliwice, a study has been undertaken on the final demolition stage. The results of this research can provide invaluable knowledge and support increasingly advanced bridge management systems. Although there are many guidebooks, standards [29,30], and sophisticated software [31] in this area, issues regarding bridge deterioration are not sufficiently recognized and described. There is still a search for better and more durable structural and material solutions. Bridge infrastructure managers are not sure of the effectiveness of the adopted maintenance strategies and repair or rehabilitation methods, for the individual components as well as the entire bridge. Moreover, the materials used for bridge construction currently differ significantly from well-recognized conventional materials. How do these materials resist corrosion after 20 or 50 years of use? It is not well understood. The builders of the bridge in Gliwice did not have this information either. They used materials and solutions that were innovative at the beginning of the 20th century. Therefore, to assess this and draw conclusions regarding the use of specific materials or new structural systems, long-term observation and research programs are necessary. During the bridge liquidation phase, a series of in-depth material and destructive tests can be carried out to determine appropriate materials for bridge construction and potentially predict any long-term problems with the current bridge. An example of this approach is the Long-Term Bridge Performance (LTBP) program initiated by the U.S. Federal Highway Agency (FHWA) [32,33]. Some of the actions are also included in the NCHRP Report (National Cooperative Highway Research Program) [34].

In the absence of any maintenance activities, an unacceptable degradation level is finally exceeded, and the bridge lifetime is reached. The deterioration can relate to selected elements (deck, abutment, bearing, and railings) or to an entire bridge. From the moment the bridge is built and put into service, the deterioration increases with wear and aging [34]. An example may be an RC (reinforced concrete) deck. After losing effective waterproofing and after the appearance of scratches or cracks, the degradation process accelerates when water with chemicals penetrate deep into the RC structure [5,8,34].

### 1.2. The River, the Region, and Impurities

Human life and activity has existed on the river banks throughout history. Rivers are a source of water for life but also were, and still are, receivers of untreated and treated sewage. Many rivers in the world are heavily polluted by industry or by human daily living needs [35,36,37,38,39,40,41,42,43,44,45,46,47,48]. Generally, the most polluted rivers were (and still are) in areas with a poor sewage management and developed heavy industry. For example, in southern Poland, Fajer and Rzetala [49] found sediments moderately to extremely contaminated with As, Cu, Co, Ni, Ba, Sr, V, and Cr, in a channel-type reservoir situated behind the weir of a water mill, in the Liswarta river. According to Tomza and Lebecka [50] in the Upper Silesian Coal Basin about 35 underground coal mines (in 2017, 30 underground coal mines) were operating. The total brines outflow from these mines was ~750,000 m^3^/day (in 2017: 700,000 m^3^/day). The total amount of salt (total dissolved solids—TDS) carried with mine water to the rivers was ~10,000 t/day, which is higher than in the ocean (up to 120 g/L). The commonest ions observed were Cl^−^ and Na^+^ with concentrations up to 70 g/L and 40 g/L, respectively. Additionally, brines typically contained several grams per liter of Ca^2+^, Mg^2+^, and SO_4_^2−^ and significant amounts of other ions, even rare ions, such as radium and barium. Barium and radium sulphates deposits occur in underground galleries. The discharge of water from several coal mines, located in the catchment area of the Odra river, have been collected and released into the river by the Olza Pipeline to protect smaller rivers from salinity, which has taken place since 1980. The radium activity has been very low in the water discharged into the Odra River via the Olza Pipeline since the early 1990s. 226Ra and 228Ra which have 10s of MBq (Megabecquerel) were released daily into the rivers along with the other mine effluents from all Polish coal mines (enhanced radium content was up to 390 kBq/m^3^ for 226Ra). The assessment of radium daily release into surface waters in 2015 into the Odra River and its tributaries showed that the Ra level did not exceed 12 MBq/day [51,52]. The poor water quality of the Odra, which flows along the western border of the Polish state, improved significantly near the end of the last century (from 1970 to 2000) [53]. This was due to the closure of numerous plants, as a result of political transformations in Poland and pro-ecological activities like the construction of sewage treatment plants [53]. However, another problem remains—the saline mine waters and many hazardous substances that for years have been accumulating in bottom sediments in rivers and reservoirs [53]. Currently, they are most likely still being released into flowing water to affect concrete structures [53]. In addition to industrial and domestic impurities, the Kłodnica river in Gliwice is contaminated with pesticides (discovered in the 1940s and used for agricultural needs after World War II) [54]. The high residue levels in the soil cause groundwater and river contamination through rain runoff. Many of these substances are persistent in the environment and are toxic, classified as a possible carcinogen and endocrine-disrupting chemicals, as well as their degradation products [54].

### 1.3. Description of the Studied Areas—The Kłodnica River and the Examined Bridge

#### 1.3.1. Kłodnica River

Kłodnica is a right-bank tributary of the Odra River. The Kłodnica drainage basin has an area of 1084.8–1125.8 km^2^ and occupies the Silesian Upland and the Racibórz Basin in southern Poland (Figure 1). It is about 75 km long with an average slope of 0.182% (1.82 m/km). The Kłodnica flows through the “GOP”—Upper Silesian Industrial District. The “GOP” is the largest industrial district of Poland, which includes the largest Silesian cities concentrated in the central part of the Silesian Voivodeship and the surrounding industrialized areas. The GOP (7500 km^2^) lies on the watershed of the Vistula and Odra river basins, which restricts the water scarcity in this area and there is not a single river that is rich in water. Along the Kłodnica’s shore, highly industrialized cities: Katowice, Ruda Śląska, Zabrze, Gliwice, and Kędzierzyn-Koźle are located. The mining industry (13 hard coal mines), metallurgy industry (15), transport industry, energy industry (more than 8 power plants), machine industry (more than 4 factories), coke industry (3 coke plants), and the chemical industry (more than 2 factories) are still being used [55,56,57,58,59,60,61,62,63,64,65,66]. Such a large quantity of factories has a great impact on the environment. This is evidenced by high pollution indicators [56,57,58,59,60,61,62,63,64,65] in rivers flowing through this area, in both their sediments and in the waters of the lakes to which they flow. Kłodnica is a submontane river with a large drop difference and significant flow variability, and has more than 12 tributaries. The bottom of the Kłodnica valley is flat and marshy. Due to high pollution on the upper Kłodnica river, the water is dark and silted. The saline waters from at least 12 mines flow into the river [66]. Thus, the water and sediment being picked up may have been contaminated many years ago. The factors, such as extensive urbanization, underground exploitation of hard coal, and much industry, have an unfavorable effect on the water quality of this river. Pollution comes from industrial and municipal sewage. Wastewater from the coke-chemical, textile, tanning, and paper industries are particularly harmful. Industrial wastewater carries large amounts of sludge, mechanical suspensions, and chemical compounds which results in the river’s inability to self-clean [55,56,57,58,59,60,61,62,63,64,65,66].

As aforementioned, life is concentrated along the river. The development of industry in the region of Upper Silesia undoubtedly influenced the quality of water in the rivers, including the Kłodnica river. Along the Kłodnica, smelting centers developed. According to Płonka [68], the foundation of the village Stara Kuźnica, in 1394, was associated with the construction of the forge (smithy). In the 1730s, a large furnace was constructed there which operated until 1786 and then was replaced with a refinery furnace. The second metallurgical center near the Kłodnica river was the village of Panewniki, where from 1650 to the end of the 17th century a forge with a bloomery and a hammer mill operated. The third smelter was in the village of Halemba and was already operating in the 15th century. In 1718, the bloomeries were replaced by a charcoal-fired blast furnace. At the beginning of the 19th century, smelting operations were discontinued in Halemba [69,70].

At the end of the 19th century, the water in the Kłodnica river was heavily polluted with industrial and municipal sewage. Dead animals were also thrown into it. The river was treated as a sewage system, at that time. For the operation of the steam tractor, a deepening of the canal was required and as a result, strongly fetid sludge was moved ashore into the center of the city. From 1919, steam transport took place along the Kłodnica. The water of the Kłodnica was slowly dying in the early part of the 1900s, evidenced by the presence of shoals of deadfish with bright spots on the skin covering the Kłodnica river. Before World War II, the transport of coke and coal to Berlin decreased from 3.2 million tons (1929) to 1.9 million tons (1936), however, the largest coal transhipment station for east Germany was still in Gliwice. Coal also flowed through the channel from Zabrze from 1795 to 1834 and then the mine water was discharged down the river until 1916. During the construction of the canal (1789–1805), bridges were also built with both stone and wooden piles. In Gliwice itself, industrial plants such as a steel mill, rolling stock repair plants, as well as a rope and wire factory, all had an impact on the environment and river. Moreover, in 1850, 1986 barges transporting coal, zinc, iron, iron products, ores, lead, sulfur, salt, explosives, and other goods left the port in Gliwice. Since reloading of these loose goods was complicated, it polluted the river, which is why Kłodnica was rich in slime. This reloading port was a large one. In the peak year of 1852, some 90,100 tons of goods, mainly coal, zinc, and iron were transported [71].

One of the main pollutants discharged into the Kłodnica river is Zn-Pb ore [72]. Most of the pollutants are carried along the tributaries and originate from several hard coal mines as underground waters, or as industrial post-flotation, or domestic sewage. Moreover, the treated sewage from municipal sewage treatment plants and discharges of household sewage from non-canalized areas flow into the Kłodnica river as well [73].

Barbusiński et al. [57] confirmed that transported suspensions contained a higher concentration of some toxic impurities compared with the amount in the bottom sediments of the Kłodnica (using mineralization with nitric acid and an atomic absorption spectrometer, with an air-acetylene flame method) (outlined in Table 1). Additionally, the metal concentrations were higher in summer [57], probably due to the lower water level in the river as a result of the evaporation of larger amounts of water at higher air temperatures. A variation in the metal concentration was also noticeable along the river length. It depended on contamination of rivers flowing into Kłodnica, caused by industrial plants located alongside them.

Government research on the composition of the Kłodnica river was completed at several points, which included locations both above and below the stream of tributaries, to determine the main sources of pollution. Some streams pretreated the river, while others clearly discharged waste. In Table 2 and Table 3, the ranges of the pollution indicators are given from the numerous research locations. This data was obtained from the Environmental Protection Inspection by the State Environmental Monitoring, using the websites of the Provincial Inspectorate for Environmental Protection in Katowice and the Chief Inspectorate for Environmental Protection. The research points and the number of markings in the Kłodnica river have changed over the years. Table 2 and Table 3 show examples of available data.

#### 1.3.2. The Examined Bridge

The understudy bridge was built around 1910 and is a classic city bridge, which had significant importance in shaping the city’s urban landscape [74]. In its long history, it survived two world wars and several major floods (Figure 2). It was also exposed to the effects of mining deformation [75]. For over 100 years, not only have the building standards have changed [76], but also the borders of the countries and institutions that manage this bridge. Unfortunately, no technical documentation has been preserved, so information about the used materials and construction methods come only from publications regarding the types of bridges built in this period and the region, as well as archival photographs (Figure 2).

Two main factors were considered when it was decided to demolish this old bridge. The first one was a too low load capacity (15 tons), which hindered the traffic in the city. The progressing deterioration of concrete of the over a hundred-year-old bridge created a risk for the introduction of further load limits. The second reason was the water flow. At high water levels in the river, the bridge repeatedly caused backwater to form and flooding of the neighboring city district (Figure 2), especially in recent years, when more and more cases of sudden and heavy rainfalls had been observed.

Figure 3 shows a side view of the bridge with its basic dimensions and structural elements. It is a three-joint RC arch with a span of 17.25 m and a width of about 20 m. The thickness of the vault is variable, with a maximum thickness of 40 cm. The reinforcement is placed only on concrete surfaces in the form of a mesh formed of 10 mm plain bars with a spacing of 30 × 30 cm. The joints are made of hewn granite blocks. The foundations were built of non-reinforced concrete and set in two rows of timber piles with a diameter of ~30 cm.

This article is a multifaceted presentation of the problem related to the technical condition of the concrete from the bridge located in Gliwice (Silesia region in the southern part of Poland) and traverses the Kłodnica river. The bridge was demolished in the summer of 2019 and an investigation of this bridge was decided because, in principle, such studies are not performed. Concrete was collected at various points in the structure and tested in the following terms.

The technical condition of concrete specimens were examined using analytical methods, comparing mainly pH and chloride ions content, as well as SEM (scanning electron microscopy) and XRD (X-ray diffraction) to answer what kind of phases were formed in concrete and what is accumulated in the river sediment.

Heavy metal content was analyzed to find out which ones had been absorbed in concrete in relation to heavy metal content in river water and bottom sediment. An analysis from a historical point of view and the impact of the river on the bridge is also undertaken.

## 2. Materials and Methods

The concrete samples were collected from accessible places and in places of various structural elements. The selection of sampling sites (closer to the water table and closer to the road surface) resulted from the source of de-icing salts and pollutants in the river.

The concrete samples with numbers 0–5, shown in Table 4, were taken from the bridge structure during its demolition. The materials were taken using a hammer and a trowel (specimens 0–4). At least 1 kg of concrete with a thickness of about 5 cm was collected from each location. Specimen 5 was prepared in the form of a 15 cm long drilled core, with a diameter equal to 10 cm. The tightly packed concrete samples were weighed and dried the same day. Then they were prepared in crushed (in a jaw crusher to grains smaller than 1 cm and in a mortar to grains smaller than 3 mm) and powdered form (in a ball mill to grains smaller than 2 μm for phases determination), according to the requirements of the test methods. The sampling locations and criteria are described in Table 4 and shown in Figure 4 (a photo of the bridge before demolition). The specific points are given in Figure 5, Figure 6, Figure 7 and Figure 8, as well as.

Liquid samples from the Kłodnica river W1 and W2 were taken in one visit at the location of the bridge (see Figure 9), in March 2020, to an amount of 2 L, and immediately tested. The W1 specimen was flowing water and the W2 specimen was water with bottom sediment. These samples were prepared for examination by isolating the water (by dynamic elution of the suspension) and sediment (dried at 40 °C to a constant mass, after filtration of the suspension). The sediments include allochthonous material (sands, silts, and gravels that arise as a result of the destruction of the bottom and the banks of the river, but also mineral and organic suspensions discharged into surface water together with surface runoff and inflow waters, as well as industrial and municipal sewage) and indigenous material (inorganic and organic substances precipitating from water, plant, and animal organisms) [56].

### 2.1. Concrete Composition

Concrete composition was tested on averaged simple samples of all the collected material. The samples at different depths were not analyzed because many years had passed and the conditions were not controlled, moreover the bridge plate was thin and moisture of the concrete depended on the weather.

The moisture of the concrete was calculated after heating in an oven at 105 °C to achieve a constant mass, according to the [78]. Then the concrete was crushed in a jaw crusher and in a mortar.

The binder content was determined by digestion of the concrete in 35% hydrochloric acid. Concrete with a grain size less than 3 mm was mixed with distilled water and acid in mass proportions: 2:1:4, respectively. Subsequently the mixture was placed on a hot plate set to 120 °C for 20 min, with sporadic stirring. As a result, the binder was completely digested (author’s method). Afterwards the samples were filtered, calcinated, the aggregate was weighed, and the binder content calculated.

### 2.2. pH Examination

The pH was measured using a portable pH-meter, 1 min after preparing the suspensions in a 1:1 (water: concrete) mass ratio. This method is not standardized. It was used to determine the difference between the samples. The determination took place directly in the suspensions, quite quickly, so that the indication came only from easily dissolved compounds. This test showed the condition of the concrete.

Another method of pH examination was prepared on the filtrates which was prepared from suspensions of 1:10 concrete: Deionized water after 1-day of dynamic leaching and filtration. This test gives lower values of pH because of using much more distilled water. Moreover, other sparingly soluble compounds dissolved after a long time. This pH were tested in the filtrates prepared for examination of salinity and metals content. In fact, there is many methods of pH measuring of the concrete [79].

### 2.3. Free Chloride Content Examination

To determine the free chloride ion content, crushed concrete was mixed with distilled water in a 1:2 mass ratio, respectively, and allowed to stand for 2 h so that the free chloride ions could release into the solution. Then the mixture was filtered and washed multiple times. Approximately 170 mL of filtrate was obtained from about 50 g of dry concrete. The free chloride ions were determined electrochemically. An ion-selective electrode (ECl-01 type by Hydromet, Gliwice, Poland), reference electrode (RL-100 type by Hydromet, Gliwice, Poland), and DC meter (MY 61 type by Mastech, Rumia, Poland) were used. The examination was made using a magnetic stirrer (MR Hei-Standard type by Heidolph Instruments GmbH & CO, Schwabach, Germany), adding a few drops of nitric acid (65%, pure for analysis) to acidify the solution to a pH value of about 1.65. The equation of the trend line prepared on the standard KCl solutions was used to calculate the concentration of free chloride ions in concrete.

### 2.4. Conductivity (Salinity) Examination

A 100 mL of deionized water with a pH of 7.6 was added to ~10 g of each crushed concrete sample and subjected to dynamic washing for 24 h. The samples were then vacuum-filtered and the salinity of the filtrates was examined. The salinity is presented in units of conductivity. The conductivity tests were carried out using a CC-505 type conductometer by Elmetron (Zabrze, Poland) and a CD-3 type conductivity cell probe by Hydromet (Gliwice, Poland). The sensor electrodes were plated with platinum, covered with platinum black to reduce the polarization phenomena occurring in highly conductive samples. The measuring range of the sensor was from 0.1 µS/cm to 10 mS/cm. For automatic temperature compensation, the set also included a CT2B-121 temperature sensor by Hydromet (Gliwice, Poland).

### 2.5. XRD (X-ray Diffraction) Examination

For phase composition analysis, an X’Pert Pro MPD X-ray diffractometer, produced by PANalytical (Westborough, MA, USA), was used. The measurements were conducted at room temperature using monochromatic Cu Kα radiation. Qualitative analysis with the support of the ICDD PDF4+ database was performed employing HighScore v4.9 software (Malvern Panalytical Ltd., Malvern, UK, 2020). The test was conducted on powdered samples. Two sample holders were used, giving a different background level in the range of 0–8 °2Th, however this has no effect on the results.

### 2.6. SEM (Scanning Electron Microscopy) Examination

The morphology of the tested sample was determined using a TESCAN Mira 3 LMU scanning electron microscope equipped with an EDS from Oxford Instruments (TESCAN, Abingdon, UK), supported by Aztec software. The specimens were carbon-coated using a Quorum Q150T ES sputter (Quorum Technologies Ltd., Guelph, ON, Canada). The tests were conducted on fractures or powdered material using the SEM–BSE operating mode. The activation energy of fluorescent radiation used for SEM-EDS analysis was 15 keV.

### 2.7. Metals Determination

Elements present in samples 0–5, given in [mg/kg] units and W1–W2, given in [ug/L] units, was diluted and determined using the following methods and equipment.

Inductively coupled plasma mass spectrometry (ICP­MS), using an Agilent 7900 ICP-MS (Agilent Technology, Santa Clara, CA, USA), was completed for Sb, As, Ba, Cr, Zn, Sn, Co, Cd, Mn, Cu, Mo, Ni, Pb, V, Fe, Mg, and Se analysis. This method was chosen because it is characterized by high sensitivity and precision with the possibility of simultaneous determination of many elements; by selectivity allowing for the determination of individual isotopes of a given element in complex matrices; and of low detection limits obtained thanks to the high ionization efficiency in plasma and by a wide range of straightness of the calibration curves. Thanks to this, both trace elements and macronutrients can be determined in one measurement. The above-mentioned method distinguishes this method from other instrumental methods used in the analysis of elements, such as ICP-OES (optical emission spectrometry) or AAS (atomic absorption spectrometry) [80].

High-performance liquid chromatography with inductively coupled plasma mass spectrometry (HPLC-ICP­MS) utilizing an Agilent 7700 Series ICP-MS with an Agilent 1260 Infinity series HPLC (Agilent Technology, Santa Clara, CA, USA) was used for the Cr (VI) study (the use of HPLC in combination with ICP-MS enabled the speciation separation of Cr (III) from Cr (VI) and the ability to determine these elements at different oxidation levels). This method was chosen because the speciation separation of Cr (III) and Cr (VI), which takes place first on the HPLC column: Cr (III) and Cr (VI) are adsorbed on the anion exchange column after the cationic Cr (III) transformation in anionic [Cr (III)—EDTA]. This allows the determination of Cr (III) and Cr (VI) separately in the ICP-MS spectrometer in the next step. The advantage of this method is the easy preparation of sewage and sludge samples for analytical determination. The HPLC-ICP-MS method is also characterized by a short analysis time, a very low detection limit (compared to other analytical methods used in the determination of Cr (VI), e.g., spectrophotometric), and the possibility of simultaneous determination of Cr (III) [81,82].

Cold-vapor atomic absorption spectroscopy (CV-AAS) utilizing a mercury analyzer, FIMS 100, Perkin Elmer, completed the Hg analysis. Mercury was determined using a CVAAS because this method uses the unique property of mercury to measure vapors at room temperature. This method, among other alternative methods for determining Hg in water and wastewater (e.g., ICP-MS or GF-AAS (graphite furnace atomic absorption spectrometry)) is distinguished by a lower limit of quantification, simple preparation of samples for analysis, the possibility of mercury stabilization during the determination, easy removal of interference, and short analysis time [82].

The dry mass of concrete was determined using a laboratory dryer (Pol-Eco-Apparatus SLW-115 Top, Wodzisław Śląski, Poland).

The samples of concrete were weighed using analytical balances (SARTORIUS, Kostrzyn Wlkp, Poland and Radwag, Radom, Poland).

The samples of suspensions were prepared using a bottle/tube roller mixer (Thermo scientific model, Thermo Fisher Scientific (China) Co., Ltd., Shanghai, China).

The samples of crushed concrete (0–5) were subjected to dynamic leaching according to [83] using a ratio of 1 mL liquid to 1 g of sample. After dynamic extraction, the leachate was left for 15 min ± 5 min. The effluent was then filtered through a 0.45 mm membrane filter using a pressure filtration device and an appropriate amount of effluent was taken to determine the concentration of the elements (ICP­MS: 2 mL; HPLC-ICP­MS: 50 μL; CV-AAS: 4 mL). Before the measurements started, calibration and current checking of devices including analysis of the control samples was completed. The correlation coefficient was greater than 0.995 for all the analyzed elements. The content of the tested elements in the individual samples was referenced against the dry mass of the concrete.

Samples W1 and W2 of flowing water and water from dynamic elution suspension before any analytical determination was filtered through a 0.45-µm membrane filter using a pressure filtration device.

The pH of the water used for dynamic elution was 6.92 which is [83]. The pH of the W1 sample was 7.94 and the pH of W2 was 7.92.

Before performing spectrometric analyzes, the test apparatus was calibrated on solutions of certified reference materials and two control samples. The analyzes of the control samples showed the accuracy of the calibration curves that were the basis for the calculations. Elemental elution calculations were performed using the LCP MHLauncher HPLC-ICP-MS and ICP-MS software and WinLab32 with a AA mercury analyzer FIMS100 [82].

## 3. Results and Discussion

### 3.1. Concrete Composition and pH of Concrete Suspensions

In Table 5, the results of pH, moisture, and “soluble phase” of concrete examination are presented. The “soluble phase” is a general name given for the part of the concrete that can be digested in hydrochloric acid (parts that are not gravel or sand). In fresh concrete, it is the hydrated cement paste content, called “binder”, but in older concrete exposed to the environment, in the definition of “soluble phase”, “corrosion products” are also included. The presence of corrosion products is confirmed by the pH values of suspensions. In healthy fresh concrete, the pH is equal to around 12.8 due to the content of Ca(OH)_2_, NaOH, KOH, and many other factors. Sodium and potassium do not form binding cement phases, they only maintain a high pH and are easily washable [10,22]. In such a case, in order to maintain a high pH, the hydrated phase decalcification process begins, i.e., their dissolution, so that free calcium ions are present in the solution in the pores of the concrete [84].

The lowest water content was found in specimen 2, which was collected from the bottom part of the arch at the very top. This section of the bridge was well ventilated. A very low pH of this sample also indicates that there was not too much original cement binder left, because of Ca(OH)_2_ dissolution and the easily soluble hydroxides were washed away. Decalcification of calcium silicate gel C-S-H means that the water would easily evaporate away because of porosity increase. Generally, it is also known that the dissolution processes create secondary pore space within the hydrated cement paste through subsequent alteration, which is observed as Ca depletion [84,85,86,87].

The reaction (pH) indicates that the concrete is not healthy. The lowest pH of the suspension was in specimens 0, 1, and 2, where the construction was exposed to the most external influences (ventilation, rainfall, exhaust emissions, and salts spraying). A slightly higher pH was observed in the concrete taken from locations 3, 4, and 5, in the foundation and at the abutment.

The binder content in healthy concrete is typically about 20% [10,22]. In tested samples, this value is not preserved. The smallest remaining amount of phase digested in HCl was in specimen 1 (9.9%), which was taken from the upper layer of the top part of the crown block. A low amount of soluble phase could be due to a high degree of concrete destruction due to leaching and reactions with de-icing salts. A very small amount of soluble phase was also in concrete taken from the abutment foundation (specimen 5), with approximately 10.5%. It is probable that heavy concrete with a high amount of aggregate was used there or phases of the binder were dissolved, as well. The concrete from the top part of the extreme arch, in specimen 3, was the least affected by the influence of the external environment. However, this concrete was the most contaminated with bituminous material, which was perceptible and visible (see Figure 10). There was less asphalt in specimen 1 than in specimen 3. Concrete in the location numbered “2” was the most exposed to evaporating water from the river. The condensate water contributed to the dissolution of the cement binder, especially calcium hydroxide. Calcium was washed out from the other hydrates and time. Moreover, carbon dioxide, from the air, bound the leached calcium to calcium carbonates (seen in Figure 5b). The concrete from the railings was made using also some black aggregate (probably basalt), where the content of soluble phases was 16.7% (partially washed off by rainwater).

### 3.2. Free Chlorides Content

Chlorides from de-icing salts cause corrosion of steel reinforcements and destruction of concrete in a dry climate when they crystallize and cause stress on pores in concrete [88,89,90,91,92,93]. According to [94], the chloride content in concrete is related to the mass of the cement and should not exceed 1% (when there are no metal elements), 0.20% or 0.40% (concrete with reinforcement and metal elements), and 0.20% (steel prestressing reinforcements). Of course, this standard was not applicable when this bridge was constructed, and are used as an indicator and guide to protect the bridge against steel corrosion and the corrosion process. So, with the consideration of the above PN-EN standard, the chloride content was greatly exceeded in the concrete of the examined bridge structure, see Table 6, Figure 11.

The significant free chloride ions content in the examined concrete is explained by the use of deicing salts over the years after construction, but the high salinity of the river also cannot be omitted. It can also explain the low content of soluble phases in the concrete. The highest mass content of chloride ions was in the specimens found under the bituminous layer sprinkled with deicing salts. In specimen 1, almost 40% of the mass of soluble phases were chlorides. This also explains the very small amount of soluble phases in specimen 1. A large number of chloride ions in sample 2 come from chlorides leaking from the above layer (specimen 1). The lowest content of chlorides was in the concrete of the abutment foundation (specimen 5) because this concrete was probably very dense, not porous, and was not constantly exposed to fresh large amounts of chloride.

The chloride ions content results suggested that the structure was also contaminated with chlorides coming from river water, and not solely from de-icing salts, due to their large amount present in the sample (Table 7). The greatest amount of chlorides was in the river water—W1 (Table 7). There are slightly less free chloride ions in the suspension—W2 (Table 7), which means, the ions are still contaminating the water.

### 3.3. pH and Conductivity (Salinity) of Filtrates

The pH values of the filtrates (prepared as 1:10 concrete: deionized water after 1-day of dynamic leaching and filtering) are summarized in Table 8. The values in Table 8 are lower than in Table 5, and cannot represent the technical condition of concrete because the concrete in the structure is not subjected to such treatment (dynamic leaching), however it can be concluded that the greatest destruction of concrete was in the area, where specimen 2 was taken. The healthiest concrete was at locations 3, 4, and 5, because even after being washed for so long with more deionized water, it kept the pH relatively high. When the reaction falls below 10, it is recommended to demolish the structure after careful and multi-criteria analysis.

The salinity of water W1 and W2 was 8610 and 8560 μS/cm, respectively, which is classified as extremely high saline water according to Ravikumar et al. [95].

The salinity of the concrete filtrates ranged from 400 to 2200 μS/cm (Figure 12). However, these results should be considered as qualitative. This measurement aims to indicate the quality of the concrete in given structural elements. That is, whether any of its component can dissolve or concrete is contaminated with ions from the external environment.

The most soluble substances were observed in samples 3, 4, and 5, which suggested a high content of easily decomposable compounds absorbed or derived from the concrete composition in the elements not directly exposed to the environment. In contrast, samples taken from the concrete railing and ceiling of the bridge (0, 1, and 2) were most exposed to leaching, so the content of salts from the environment and easily soluble hydrates of the cement paste was relatively low.

From the salinity (Figure 12), the chloride content, and pH of the structural elements of concrete it can be concluded that samples 0, 1, and 2 had a low salinity, and these were mainly salty with chlorides, and the cement binder was dissolved and washed away, which means that the concrete in these structural elements was mechanically weak. Sample 1 (concrete under asphalt), was the most contaminated with chlorides. Whereas, in samples 3, 4, and in particular 5, there was a certain amount of easily soluble not binding phases. These phases were detected by XRD method (in the next section).

The pH of the concrete railings was lower than that of healthy concrete, and it is explained by the exposure of railings to atmospheric precipitation, chlorides ingress, and carbonation reactions. The pH of this concrete was not sufficient to protect steel reinforcement against corrosion. This is very notable in the image shown in Figure 8, where corroded bars revealed and where the sample “0” is seen, as well.

### 3.4. XRD

Besides sodium and potassium hydroxides, calcium hydroxide (Portlandite—in crystalline form) is easily soluble. It is customary that the presence of portlandite in hydrated cement paste shows that the concrete can self-heal. Looking at the pH, it was hoped this compound would be detected in trace amounts in sample 3 or 4, maybe in 5, but it was not detected anywhere, as shown in Figure 13 and Table 9. The number of counts in the graph (Figure 13) relates to the diffractogram of sample 5.

The XRD patterns of specimens 0, 1, 2, 3, 4, and 5 (Figure 13) show the presence of aggregates and calcium carbonates. This means the concrete was heavily carbonized, to the extent that not one cement phase could be detected. The biggest surprise was the lack of sulphate phases indicating corrosion of the concrete. The corrosive phase, in addition to calcium carbonates, was hydrocalumite, present in specimens 3 and 4, called Friedel’s salt. This chloride phase is unstable when the pH drops below 12. Among the discovered calcium carbonates: Calcite was present ubiquitously and vaterite was present in specimens 0, 1, 2, and 3—in the upper sections of the construction where carbon dioxide was more available as CO_2_ or HCO_3_^−^ ions. Calcite may be a part of the aggregate, but there it was mainly a product of concrete destruction. Vaterite, in turn, recrystallizes to form calcite or aragonite but can also be derived from biochemical sediments [96]. After all, it cannot be ruled out that the concrete has also been biodeteriorated [97]. All the detected phases in the individual samples are summarized in Table 9.

The following crystalline phases were observed in XRD patterns: Quartz, albite (silicate from the group of plagioclases; calcite and kaolinite are the products of albite weathering), kalsilite (aluminosilicate, an aggregate component), anorthite (a rare plagioclase that forms mixed crystals with albite), illite (common clay mineral; weathering product of feldspar and other aluminosilicates, an aggregate component), kaolinite (common clay mineral; weathering product of feldspar and other aluminosilicates, an aggregate component), hydrotalcite (hydrated basic aluminum magnesium carbonate which may also be a product of corrosion as it is the not very widespread hydrated carbonate of the low-temperature weathering zone), microcline (potassium feldspar, an aggregate component), rutile (a mineral of the oxide cluster, a very common, widespread component of many rocks, although some component of the TiO_2_ is believed to originate from river pollutants), gismondine (a mineral from the group of silicates, belongs to the zeolite group and is a component of blast furnace slag, so it may be from river pollution), phlogopite (a common rock-forming mineral from the silicate group, included in the group of mica, a product of hydrothermal processes, it is formed in the voids of volcanic rocks such as in basalt lava and is an aggregate component), and clinochlore (a mineral from the aluminosilicates magnesium chlorite group). In specimens 1 to 4, the composition of concrete was expected to be the same, however this is not the case. This is due to the aggregates in the concrete being weathered. Most of the aggregates probably came from Lower Silesia [96].

### 3.5. SEM

In the crushed concrete, taken from specimen 4, Cu was detected, probably in metallic form, as shown in Figure 14.

Observations and examinations of the elemental composition of concrete showed a large amount of quartz and calcium carbonate in the amorphous and cuboids crystal region, see Figure 15. In the amorphous phase of the concrete, hydrated calcium aluminosilicate C-A-S-H and calcium silicate C-S-H phases were observable, along with aluminosilicates such as sodium aluminosilicate.

An accumulation of needle-like sulfate salts, possibly ettringite Ca_6_Al_2_(SO_4_)_3_(OH)_12_·26H_2_O or even thaumasite: Ca_3_Si(OH)_6_(CO_3_)(SO_4_)·12H_2_O phases, were observed in the pores, located here in a small amount, and not detected using diffractometry, however, they are seen in the SEM micrographs in Figure 16.

Observation of W2 sediment from the river indicated it was not composed of round gravels nor sands typical for river debris, but it rather formed dust and ashes with both oxidized and hydrated phases, or even slag polluting the river, as shown in Figure 17.

The detected Zn, Co, Cu, Ti, P, and Na elements were uniformly distributed throughout the mass, but the rest of the elements were not uniformly distributed, they formed clusters, interlayered in the minerals, and pointed to mineralogical separateness, as seen in Figure 18.

Inclusions in the sludge may indicate the presence of iron ore but could also correspond to other compositions from the underground mine waters. Moreover, the zinc and lead ore deposits in the Bytom trough co-exist with the ore-bearing dolomite series of the middle Triassic.

### 3.6. Elements Determination

Table 10 contains the results of concentration of the elements in filtered water from the Kłodnica river (W1) and in the water after dynamic leaching and filtering of the suspension at the bottom of the Kłodnica river (W2). Table 11, in turn, presents results after dynamic leaching of the concrete’s elements taken from the structural elements of the bridge.

Most elements have a comparable concentration in the flowing water and the water from dynamic leaching of the river water with suspension. This means that some elements were absorbed in the mineral suspension, while others were present in the free form of the water. Vanadium, copper, manganese, zinc, chromium (VI), and cadmium undoubtedly belong to the elements that were absorbed more easily into the suspension. However, lead seemed to be more difficult to absorb into the bottom sediment, therefore their concentration was higher in the flowing water than in water with suspension after dynamic leaching, which has also been previously noted by Barbusiński et al. [56,57].

The concentration of chromium, nickel, lead, iron, and cadmium was lower in the river, as shown in the data from Table 2 and Table 3. However, there was more magnesium. The amounts of barium, manganese, and zinc in the Kłodnica river’s water were also significant, greater than in typical river water.

The highest content of magnesium, cobalt, vanadium, arsenic, and chromium was observed in specimen 2. The highest content of nickel was in specimen 5. The highest molybdenum content was in specimen 0. Antimony and chromium content was the highest in specimens 0, 1, and 2. In specimens 0 and 5, the copper content was greater than in the other specimens. The most barium and zinc was seen in specimen 3. The least chromium (VI) was in specimens 3 and 4, and the greatest amount in specimens 0 and 2. The above concrete elemental tests indicate that the most metal was absorbed by the concrete just above the main river current (specimen 2), which indicates the transfer of elements due to the evaporation and condensation of the river water. There is also an assumption that the metals found in the bridge railing may also come from car exhaust fumes, but this has not been proven.

The magnesium concentration in the river over the years up to the current year showed it is still high and was also high in the concrete, in particular, seen in sample 2, 1, and 0. Nickel was mainly observed in the foundations of the bridge (specimen 5), for unknown reasons, but they may even be related to the composition of the concrete or to core extraction. Such is also the case with chromium and chromium VI, which accumulated in the concrete after many years of contact with the river. There was considerable arsenic found in the water, as evidenced by the content in sample 2. Many years back, significant amounts of copper were observed in the river water (Table 2 and Table 3) and now its content has dropped considerably. However, Cu accumulated in the concrete, due to floods flooding the bridge. Therefore, concrete tests, especially on samples 0 and 5, demonstrated an elevated copper content from many years ago in the water of the Kłodnica river. Iron was not absorbed by the concrete. Mercury, manganese, zinc, and lead were also not readily absorbed by the concrete. The analysis is quite complicated as it depends on the ability of these elements to accumulate in the concrete’s mass.

## 4. General Discussion

A lot of destructive factors have influenced the structure of the bridge, therefore all of them decreased its technical properties. For example, within dissolved rain water, exhaust gases and de-icing salts flowed off the bridge flooring and lowered the pH of concrete by Cl^−^ and HCO_3_^−^ ions diffusion into the concrete structure and OH^−^ ions released out of the concrete. As a consequence, calcium phases were unstable in the reduced pH and dissolved, but under these conditions calcium carbonates were formed. A chloride phase was also found, the so-called Friedel’s salt, which, due to the large amount of chloride ions, most likely arose from the original ettringite. The gypsum was not found, because the pH of the concrete would have to drop to 8 [10,22,84,85,86]. In the most degraded concrete specimens, the greatest amounts of metals were also detected, in particular Mg, Co, V, Mo, Sb, Cr, Cu, Cr (VI), and As. It was believed that the most metals from the river water would be in sample 4, because it came from the lowest element of the bridge, situated the closest to the river, although the concrete from this part was not influenced so strongly by river water, probably, judging by the pH and low chloride content, there the concrete was quite tight.

The relatively high nickel and copper content in the foundations 1.5 m below the bottom of the river was surprising. Comparing the content of manganese to iron according to SEM analysis and the tests in water samples W1 and W2, it can be concluded that there was less manganese in the solid phase and less iron in the liquid phase. Fe, Mn, Hg, Pb, Cd, and Sn were on the verge of detection in concrete, while in water specimens, there was a great amount of Mn. Barium was not found in the river sediment, but there was a considerable amount in the river water, and it seemed to be well absorbed in the concrete. There was quite a lot of zinc in the bottom sediment and was easily washable. In concrete, Zn was mostly found in the foundation under the river bed. This may indicate that it was a very old contamination that had found its way into the ground. The quite high content of chromium VI in concrete is surprising, so it was absorbed since its traces are still present in sediment and river water.

In order to carry out precise analyzes for the presence of metals in concrete, additional testing methods, such as TEM (transmission electron microscopy) or XPS (x-ray photoelectron spectroscopy) should be employed. At this stage, it only can be concluded which metals have been absorbed by the concrete, and the amount can be compared with the presence in water and sediment. This analysis for the presence of metals in the bridge concrete is not easy and cannot be referred to in the literature data on the topic in the field of fresh concrete research. It is known that metals do not absorb to the same degree in concrete [98,99,100,101].

Heavy industry may cause significant pollution of the natural environment due to the release of wastewaters. This phenomenon is well known in the Upper Silesian Coal Basin where the underground exploitation of coal, oil, and gas or other resources has taken place [51,52,53,54,55,56,57,58,59,60,61,62,63,64]. The issue of water quality is very important, particularly in Poland, which has poor water resources compared to other European countries. Currently, the amount of pollution is reducing due to favorable changes in the output from wastewater treatment plants, but also because of the elimination of a majority of arduous industrial plants, over the last 30 years in the Kłodnica river [53]. However, the pollution found in the river today is still an effect of policy over the past decades. Industrial waters with considerable amounts of slime, salt, and some toxic metals supplied the Kłodnica river for many years [56,57,63]. This has a large impact on the lack of usable water resources, and on the landscape, agriculture, and recreation, but the impact of a polluted river on the engineering structures that traverse the river is also evident. River pollution affects building structures as well.

## 5. Conclusions and Summary

The examined concrete from the 100-year-old bridge over the heavily polluted river was in a very poor condition. By comparing the content of heavy metals in the river, in the bottom sediment and in concrete, it was proved that concrete did not absorb all types of metals, of which some metal were also found that were no longer present in the river. Thus, precise concrete tests can indicate historical events and the state of the environment from years ago.

The paper presented unique tests of a bridge undergoing demolition. The novelty of the article is the fact that such tests, when the bridges are liquidated, have not been carried out in that way. Moreover, it is not possible to collect samples from working and old bridges, from such places, because the structure would be damaged. Therefore, the results of these studies can be applied to other similar concrete bridges that are located over polluted rivers in a similar climate.

The upper elements of the bridge (concrete railings, crown blocks, and top part of extremity of an arch) were the most carbonated, proven by the detection of the presence of calcium carbonates. Moreover, chloride ions were also present in a great amount. These bridge elements had the most frequent contact with the exhaust gases. The lowest pH in the concrete came from concrete railings and crown blocks.

The penetration of chloride ions was mainly attributed to the corrosion of steel reinforcement, although this work does not deal with this topic, nor with the traditional testing of the bridge using mechanical methods. However from the results of concrete tests, it could be concluded that the reinforcement was corroded, and the structure was also already weak.

Even though the bridge was soaked with highly saline river water, sulphate phases were probably not detected in the concrete, but a small amount of a typical chlorine phase, a Friedels’ salt, was detected. This shows that the concrete was exposed to such significant deterioration that even the corrosive phases dissolved. The high degree of concrete deterioration was also confirmed by the weathering of the aggregate.

Therefore, comparing the influence of heavy metals and the salinity of the river, it should be stated that for the salinity: Chlorides and carbonate ions greatly influenced the corrosion of the reinforced concrete of the bridge, while the heavy metals found in the concrete tell us only what else the river was contaminated with, years ago.

The leaching of crushed concrete examination showed the presence of many elements like magnesium, cobalt, vanadium, arsenic, nickel, molybdenum, antimony, chromium, chromium (VI), copper, barium, and zinc. Other heavy metals were absorbed in very small quantities. Currently, some elements were found in trace amounts in the river water, but their presence was visible in the concrete, such as chromium, copper, and arsenic. Iron, zinc, lead, magnesium, manganese, and barium were in a significant amount in the river, but these elements did not absorb into the concrete in any appreciable amount. So it is possible to state which elements, found in the Kłodnica river, originated from a time when studies of water analysis were not conducted.

The scope of research presented in the article is not customary because the authors wanted to discover something more. In addition to research on the technical condition of concrete, it was puzzling to find elements in the concrete that would testify to the contamination of river water from years ago. Due to the fact that the metals detected in concrete did not affect, or their influence on the technical condition of the structure is not known, these discoveries could be considered as an extension of the historical knowledge of ecologists on the pollution of water courses in industrial agglomerations.

Undoubtedly, many factors acting synergistically contributed to the reduction of the life of the examined bridge. However, it cannot be said which of these factors was decisive. It is a very individual matter that depends on the location of the bridge, load, material quality, operations, maintenance strategy, and many more. In this case, however, the polluted river could have had a greater impact than the load, because the load is the same on the neighboring bridges in the area (not demolished, not located above the Kłodnica river).

The results of the research and used methodology could be applied in the future to broaden knowledge about the durability and deterioration processes of bridges. It may allow us to extend the life cycle of old bridges and continue their safe operation, whilst giving critical information needed to decide about decommissioning and replacing it with a new one. New knowledge about these phenomena will also help to develop better methods for designing and modeling new bridges. In addition, it also aids in understanding the durability and predicting safety risks, using advanced maintenance decision-making tools such as the digital information management method (the BIM—Building Information Modeling).

In summary, the concrete of bridges over saline rivers quickly dissolves and decomposes because it works in an environment where the occurrence of chemical reactions is facilitated. Therefore, in order to better explain the technical condition of the structures, we cannot neglect the recognition of the surrounding environment. The ground, the air, and the water should be examined because all environmental factors (natural and anthropogenic) affect the condition of the structure.

## Figures and Tables

**Figure 1 materials-14-00981-f001:**
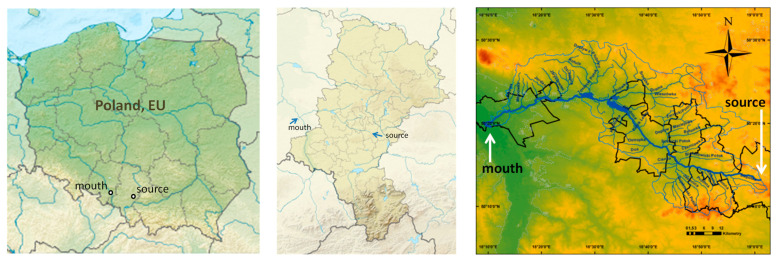
Maps showing the Kłodnica river basin [66,67].

**Figure 2 materials-14-00981-f002:**
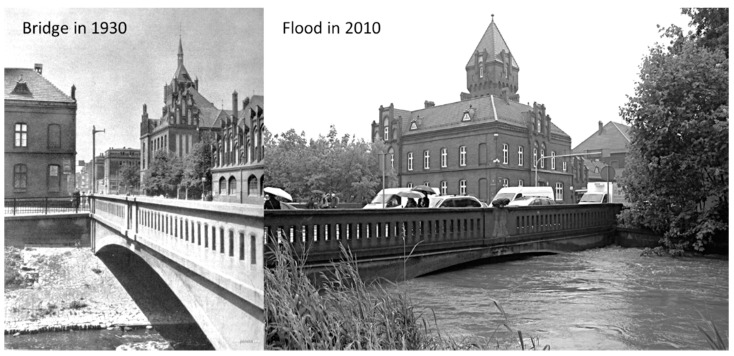
View of the bridge in 1930 and during a flood in 2010 [77].

**Figure 3 materials-14-00981-f003:**
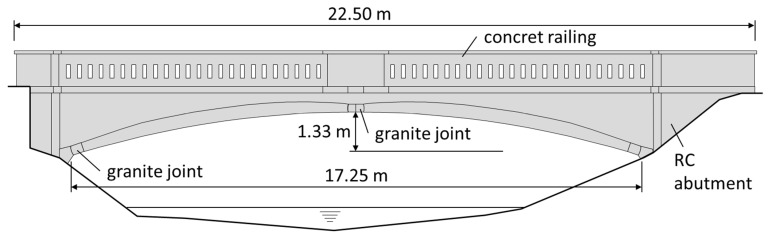
Side view of the bridge.

**Figure 4 materials-14-00981-f004:**
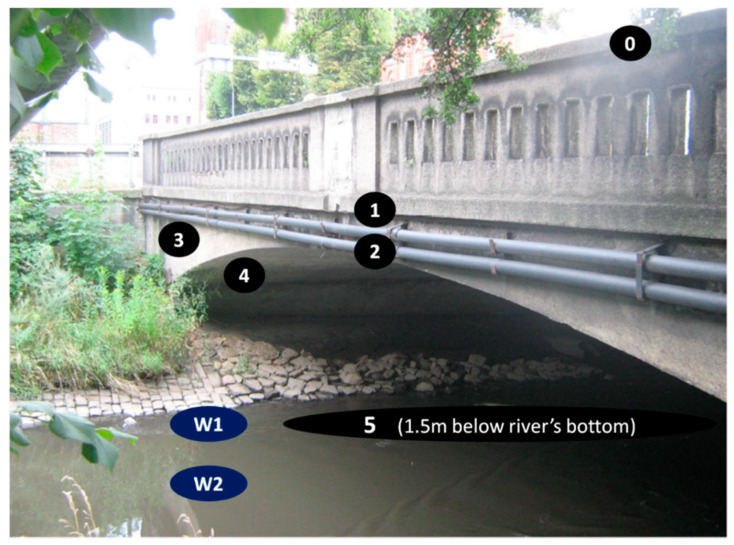
Location of samples of concrete and water shown before demolition of the bridge.

**Figure 5 materials-14-00981-f005:**
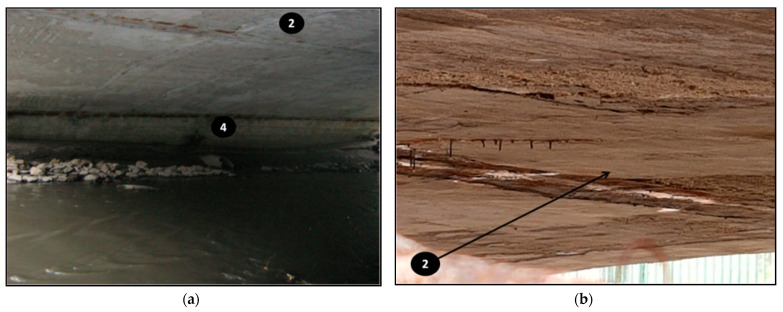
The sampling sites during demolition: Corroded steel mesh and carbonate icicles: (**a**) location of samples 2 and 4, (**b**) location of samples 2, (**c**) location of samples 1 and 2, (**d**) location of samples 3 and 4.

**Figure 6 materials-14-00981-f006:**
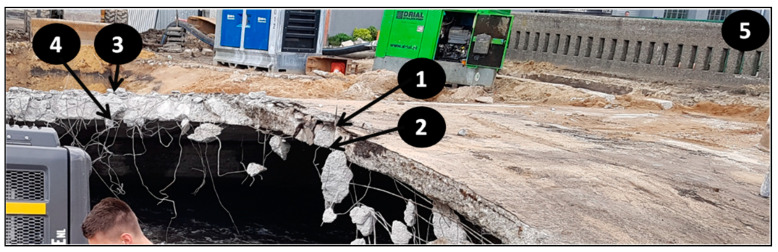
The RC (reinforced concrete) slab after removing the bituminous layer from the bridge.

**Figure 7 materials-14-00981-f007:**
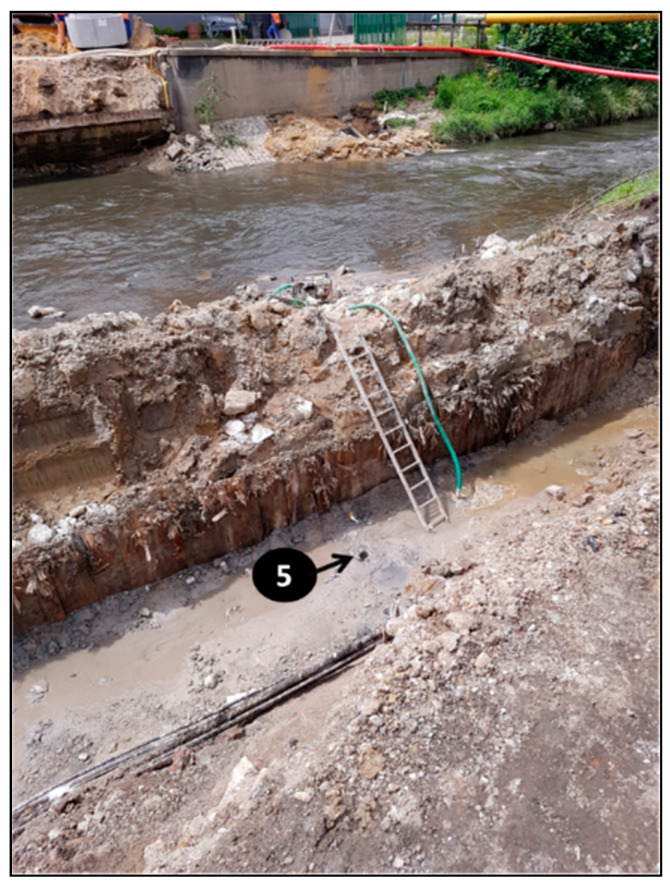
Location of foundation borehole.

**Figure 8 materials-14-00981-f008:**
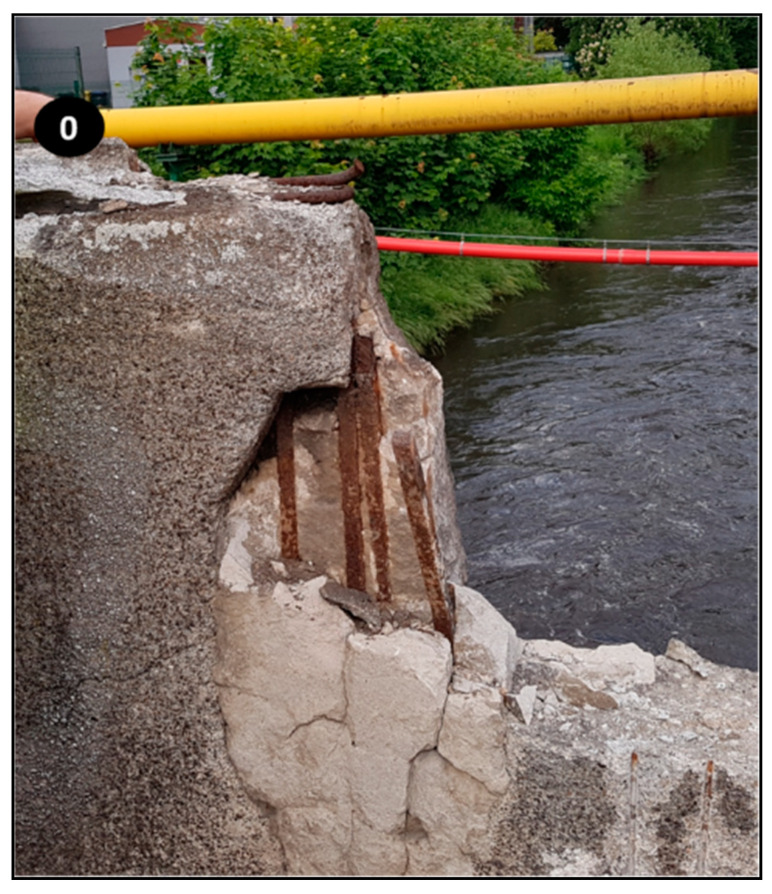
Corroded steel bars in the concrete of the balustrade.

**Figure 9 materials-14-00981-f009:**
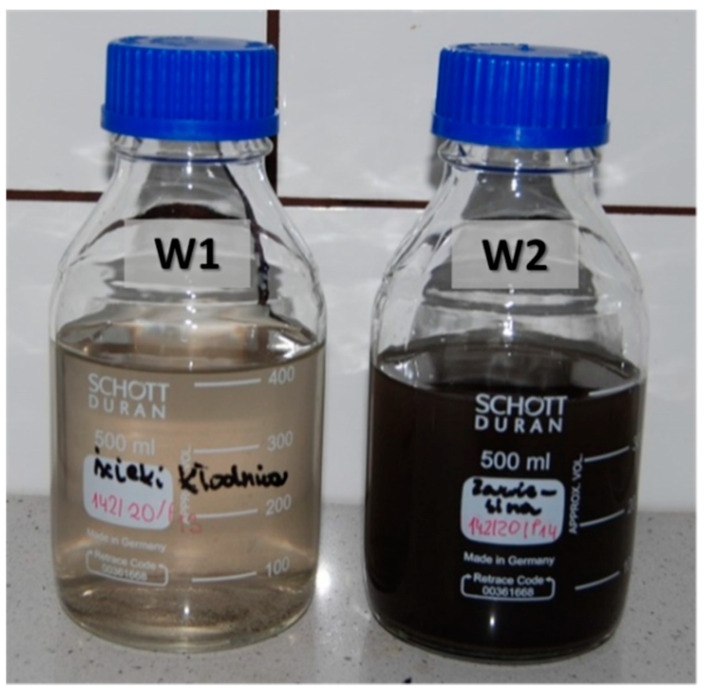
Water and suspension samples taken in March 2020 from the Kłodnica river; flowing water (W1) and water collected at the bottom of the river (W2).

**Figure 10 materials-14-00981-f010:**
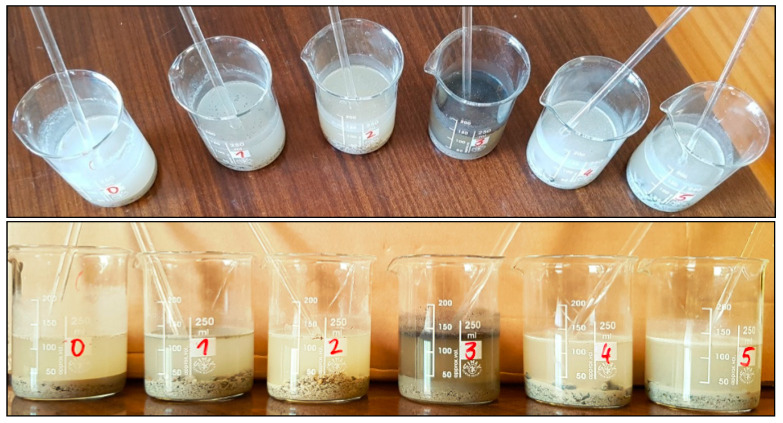
Suspensions made from deionized water and crushed concrete from the bridge.

**Figure 11 materials-14-00981-f011:**
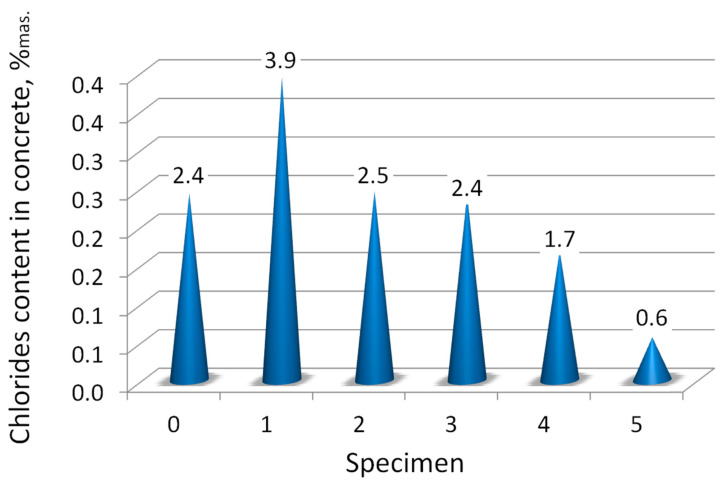
The free chloride ions content in concrete samples from the bridge.

**Figure 12 materials-14-00981-f012:**
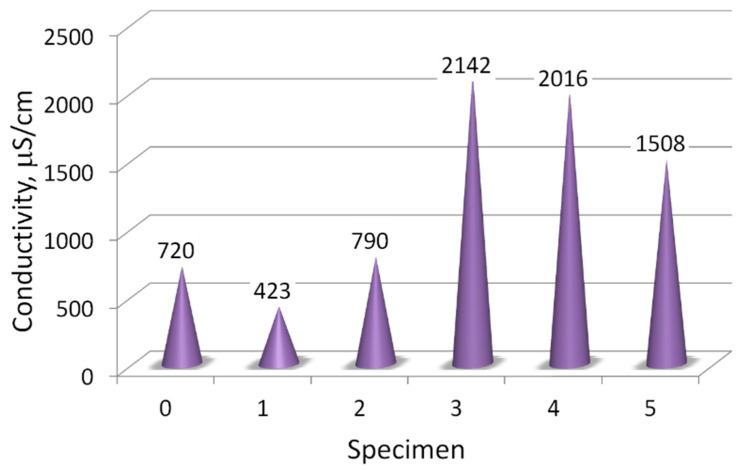
The conductivity of filtrates made of concrete samples from the bridge.

**Figure 13 materials-14-00981-f013:**
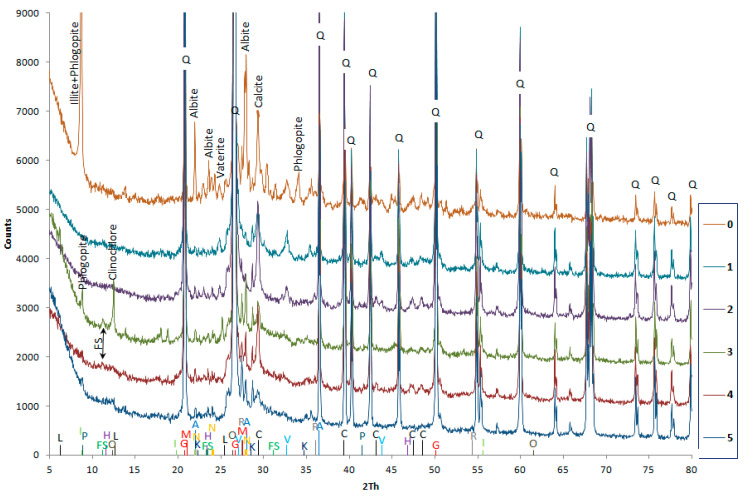
The XRD patterns of specimens 0 to 5.

**Figure 14 materials-14-00981-f014:**
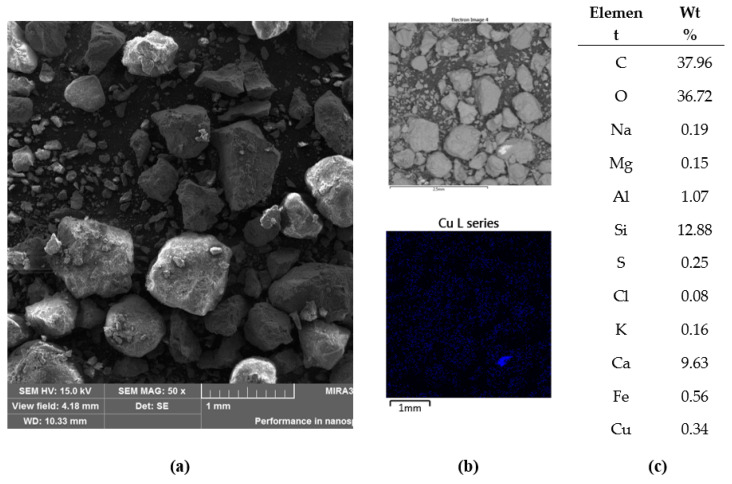
Crushed concrete on carbon tape, specimen 4: (**a**) SEM-SE image; (**b**) SEM-BSE image, with a map of Cu location, and (**c**) the elemental content over the whole shown area.

**Figure 15 materials-14-00981-f015:**
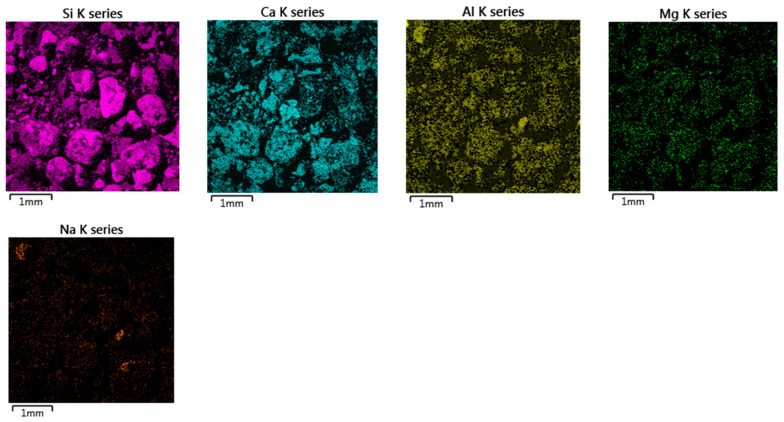
Crushed concrete on carbon tape, specimen 4: Si, Ca, Al, Mg, and Na maps over the analyzed area shown in Figure 14b. Mg is evenly distributed throughout the image, suggesting it is absorbed freely.

**Figure 16 materials-14-00981-f016:**
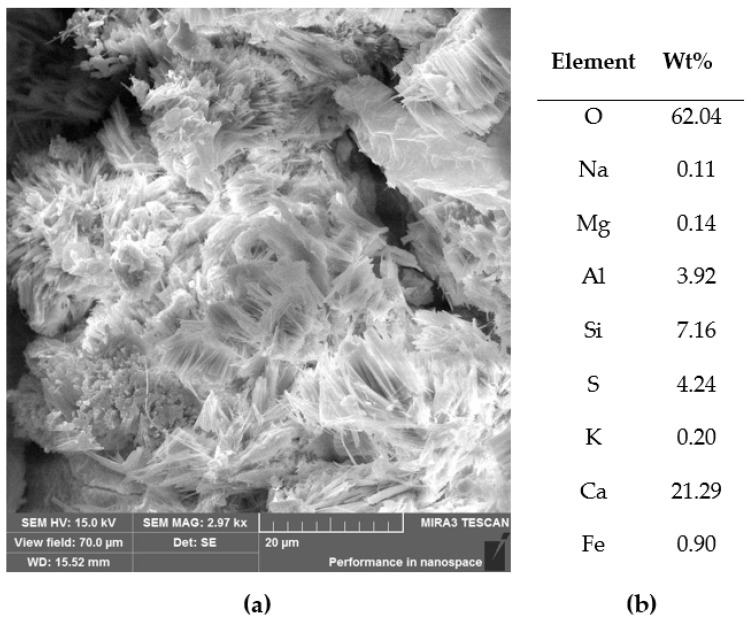
A crystalline form from inside a pore of the concrete, specimen 4: (**a**) SEM-SE image and (**b**) elemental content of the entire shown area.

**Figure 17 materials-14-00981-f017:**
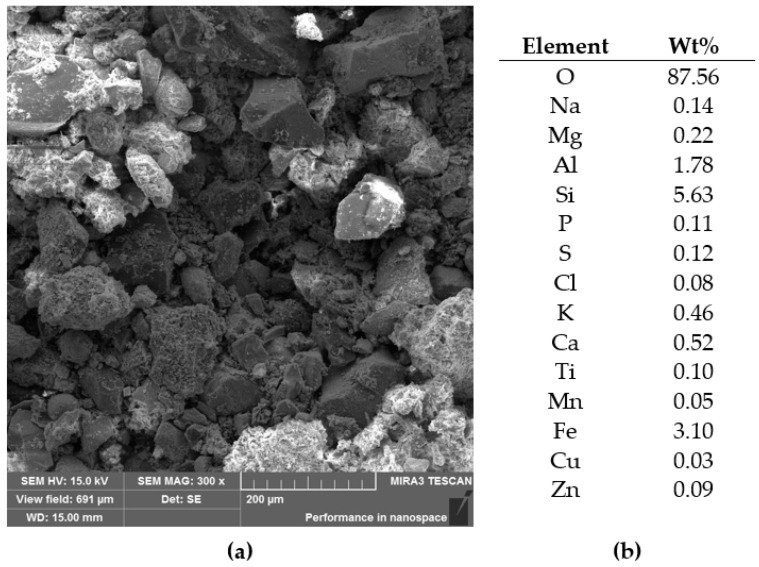
Sediment from the Kłodnica river, from specimen W2: (**a**) SEM-SE image and (**b**) elemental content from the whole shown area.

**Figure 18 materials-14-00981-f018:**
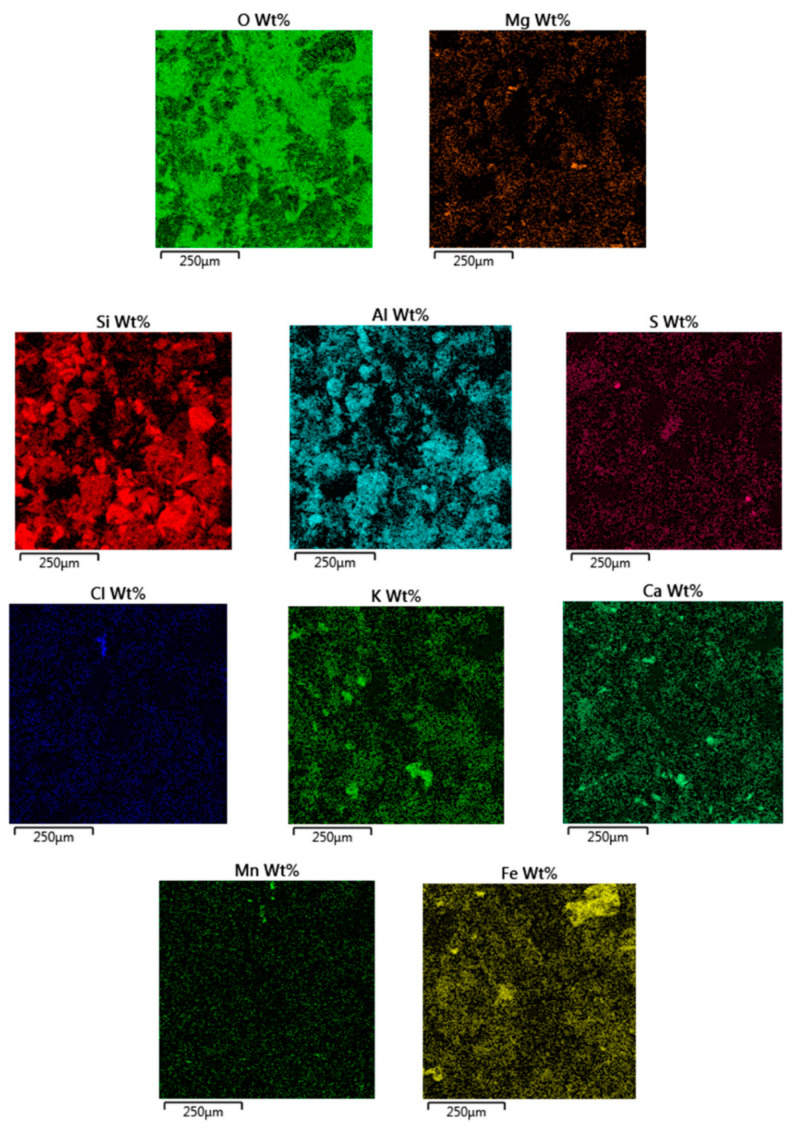
Microanalysis of specimen W2 sediment from Kłodnica river: Elemental content maps over the area given in Figure 17a.

**Table 1 materials-14-00981-t001:** A comparison of heavy metal content compared to the dry mass of suspensions in bottom sediments and suspensions in the Kłodnica taken at eight measuring points [56,57].

Metal	Metals in Bottom Sediments (Min–Max) in 2011–2012 (mg/kg)	Metals in Suspensions (Min–Max) in 2011–2012 (mg/kg)	Metals in Suspensions (Max) at the End of the 90s (mg/kg)
Zn	35–500	<6610	11,500
Pb	7–80	4–599	742
Cu	1–46	>100	2462
Ni	1–30	3–74	3317
Cr	1–24	<40	1613
Cd	1.3–8.1	0.2–72.9	120
Fe	<1000	720–6300	385,072
Mn	10–1000	13–9600	22,107

**Table 2 materials-14-00981-t002:** Average values of selected pollution indicators in the Kłodnica river from 1992–2018, the ranges are based on several measurement points. Based on data from the Environmental Protection Inspection obtained by the State Environmental Monitoring.

Year	Chlorides (mg Cl^−^/L)	Sulphates (mg SO_4_^2−^/L)	Solute (mg/L)	Mg (mg/L)	Cr (mg/L)	Cu (mg/L)	Pb (mg/L)	Fe (mg/L)
1992	1117–2647	213–432	2588–3965	nd	nd	0.01	0.01–0.02	nd
2002	904–2107	221–597	2436–4966	50–103	0.01	0.01–0.015	0.042	0.292–0.874
2003	1700–3200	300–700	3800–7000	nd	nd	nd	nd	nd
2004	1100–2500	250–630	2700–5700	nd	nd	nd	nd	nd
2005	2920–3430	360–730	6240–7460	129–159	0.01	0.01	0.002	0.129–0.275
2006	986–2266	160–492	2270–5170	nd	0.01	0.01	0.01–0.04	0.11–0.18
2007	1713–1890	250–600	4575	99	0.003	0.004	0.001	0.1–0.3
2008	1397–1787	247–552	3099–4406	59–94	0.015	0.02	nd	nd
2012	1824–2039	410–640	4320–3890	nd	<0.01	0.054	nd	nd
2018	883–1904	289–582	2109–5073	nd	nd	0.004	nd	nd

nd—not detected.

**Table 3 materials-14-00981-t003:** Average values of specific indicators and their compounds from 2016–2018. Based on data from the Environmental Protection Inspection obtained by the State Environmental Monitoring.

Year	Cu (mg/L)	Cd (μg/L)	Pb (μg/L)	Ni (μg/L)
2016	0.01–0.012	0.15–0.39	0.6–2.1	3.9–4.1
2017	nd	0.15	1.09–2.28	5.28–5.40
2018	0.004	0.07–0.34	1.2–2.1	3.0–3.1

nd—not detected.

**Table 4 materials-14-00981-t004:** Characterization of samples.

Sample	Location of the Sample	Comments/Criteria
0	Concrete railings	Exposed to the influence of the urban environment, road transport, river condensate
1	Top part of the crown block	Exposed to river condensates but mainly to salt and rainwater solutions leaking asphalt and exposed to internal components (e.g., bitumen from insulation)
2	Bottom part of the crown block	Most exposed to vapors
3	Top part of the extremity of an arch	Exposed to corrosion in the least way
4	Bottom part of the extremity of an arch	Exposed to vapors and freezing
5	Drilled from foundations 1.5 m below the bottom of the river	About 1 m below the ordinate of the river bottom, exposed to groundwater
W1	Flowing water from the river	Collected at the riverbank, upstream
W2	Water with sediment from the river taken at the river bottom	Samples after dynamic elution of suspension and filtration: (a) Water; (b) dried at 40 °C to a constant sediment mass

**Table 5 materials-14-00981-t005:** A range of parameters for the concrete specimens.

Specimen	Moisture, %_mas._	pH	Soluble Phase, %_mas._
0	5.3	11.00	16.7
1	5.1	10.62	9.9
2	2.8	10.51	12.1
3	5.9	12.00	17.6
4	6.2	11.94	12.0
5	5.0	11.72	10.5

**Table 6 materials-14-00981-t006:** The free chloride ions content in the concrete and in digested in HCl phase.

Specimen	Free Chlorides Content in Concrete, %_mas._	Free Chlorides Content in Soluble Phases, %_mas._
0	2.43	14.58
1	3.93	39.58
2	2.46	20.22
3	2.41	13.70
4	1.68	14.03
5	0.56	5.33

**Table 7 materials-14-00981-t007:** The chloride content in the water of the Kłodnica river.

Specimen	Chlorides Content, mg/L
W1	3547
W2	2995

**Table 8 materials-14-00981-t008:** The pH of the river water and concrete filtrates after 24 h of dynamic leaching.

Specimen	0	1	2	3	4	5	W1	W2
pH	11.05	9.95	7.13	12.03	12.04	11.91	7.67	7.92

**Table 9 materials-14-00981-t009:** The crystalline phases in concrete specimens of the bridge.

Sample No.	SiO_2_	CaCO_3_	CaCO_3_	(Na,Ca)Al(Si,Al)_3_O_8_	KAl(SiO_4_)	Al_2_CaO_8_Si_2_	Al_3_H_2_KO_12_Si_3_	Al_2_H_4_O_9_Si_2_	Al_0.33_C_0.165_H_2.96_Mg_0.67_O_2.975_	AlKO_8_Si_3_	TiO_2_	CaAl_2_Si_2_O_8_.4H_2_O	KMg_3_(Si_3_Al)O_10_(F,OH)_2_	(Mg,Fe)_6_(Si,Al)_4_O_10_(OH)_8_	Ca_4_Al_2_O_6_Cl_2_.10H_2_O
	Q—Quartz	C—Calcite	V—Vaterite	A—Albite	K—Kalsilite	N—Anorthite	I—Illite	O—Kaolinite	H—Hydrotalcite	M—Microcline	R—Rutile	G—Gismondine	P—Phlogopite	L—Clinochlore	FS—Hydrocalumite
0	√	√	√	√	√		√						√		
1	√	√	√		√	√									
2	√	√	√		√	√	√	√		√	√	√			
3	√	√	√	√	√	√	√					√	√	√	√
4	√	√			√	√		√		√	√				√
5	√	√		√	√	√	√	√	√	√					

**Table 10 materials-14-00981-t010:** The total metal content (μg/L) in water W1 and from the filtered eluted suspension W2 from the Kłodnica river.

Metal Contents	Specimen W1 x_av_ ± s_x_	Specimen W2 x_av_ ± s_x_
Mg	151973 ± 4209	150631 ± 6780
Co	1.088 ± 0.040	1.159 ± 0.037
Se	2.185 ± 0.829	2.125 ± 1.045
Sn	0.013 ± 0.020	0.003 ± 0.001
V	±0.072	0.607 ± 0.067
Ni	<1	<1
Mo	2.313 ± 0.153	2.237 ± 0.173
Sb	1.784 ± 0.104	2.221 ± 0.084
Cr	<1	<1
Cu	1.003 ± 0.392	3.217 ± 0.274
Ba	61.256 ± 7.287	61.579 ± 8.677
Hg	<0.0075	<0.001
Mn	293.3 ± 26.7	411.9 ± 11.9
Zn	4.785 ± 0.134	7.799 ± 1.713
Pb	0.022 ± 0.001	<0.001
Cr (VI)	<0.075	<0.091
Fe	<1	<1
As	1.161 ± 0.147	1.057 ± 0.255
Cd	0.009 ± 0.010	0.029 ± 0.006

Symbols: x_av_—the average value of the test result; s_x_—standard deviation determining the dispersion of results obtained for replicates of the analyzed sample from the average value.

**Table 11 materials-14-00981-t011:** The total metal content in the leachants of the concrete specimens taken from the bridge above the Kłodnica river in Gliwice, in mg/kg.

Metal Contents	Specimen 0 x_av_ ± s_x_	Specimen 1 x_av_ ± s_x_	Specimen 2 x_av_ ± s_x_	Specimen 3 x_av_ ± s_x_	Specimen 4 x_av_ ± s_x_	Specimen 5 x_av_ ± s_x_
Mg	0.133 ± 0.011	0.367 ± 0.009	1.114 ± 0.042	0.035 ± 0.042	0.015 ± 0.007	0.030 ± 0.013
Co	<0.001	<0.001	0.008 ± 0.001	<0.001	<0.001	0.003 ± 0.001
Se	0.004 ± 0.002	0.002 ± 0.001	0.003 ± 0.001	0.004 ± 0.002	0.002 ± 0.001	0.003 ± 0.001
Sn	0.002 ± 0.001	0.002 ± 0.001	<0.001	<0.001	<0.001	<0.002
V	<0.001	0.017 ± 0.009	0.061 ± 0.002	<0.001	<0.001	<0.001
Ni	<0.001	<0.001	0.007 ± 0.001	0.005 ± 0.001	<0.001	0.097 ± 0.001
Mo	0.144 ± 0.003	0.007 ± 0.001	0.007 ± 0.001	0.003 ± 0.001	0.002 ± 0.001	0.003 ± 0.001
Sb	0.005 ± 0.002	0.005 ± 0.002	0.005 ± 0.001	0.002 ± 0.001	0.002 ± 0.001	0.002 ± 0.001
Cr	0.191 ± 0.001	0.116 ± 0.001	0.210 ± 0.005	0.035 ± 0.004	0.075 ± 0.002	0.048 ± 0.002
Cu	0.474 ± 0.031	0.240 ± 0.014	0.086 ± 0.004	0.125 ± 0.008	0.022 ± 0.011	0.591 ± 0.003
Ba	0.032 ± 0.002	0.045 ± 0.001	0.060 ± 0.002	0.187 ± 0.010	0.036 ± 0.019	0.060 ± 0.002
Hg	<0.001	<0.001	<0.001	<0.001	<0.001	<0.001
Mn	<0.001	<0.001	<0.001	<0.001	<0.001	<0.001
Zn	<0.001	<0.001	<0.001	0.003 ± 0.001	<0.001	0.007
Pb	<0.001	<0.001	<0.001	<0.001	<0.001	<0.001
Cr (VI)	0.176 ± 0.004	0.097 ± 0.001	0.207 ± 0.008	0.024 ± 0.001	0.013 ± 0.009	0.060 ± 0.001
Fe	<0.001	<0.001	<0.001	<0.001	<0.001	<0.001
As	0.004 ± 0.001	0.021 ± 0.003	0.039 ± 0.001	0.004 ± 0.001	<0.001	0.004 ± 0.001
Cd	<0.001	<0.001	<0.001	<0.001	<0.001	<0.001

Symbols: x_av_—the average value of the test result; s_x_—standard deviation determining the dispersion of results obtained for replicates of the analyzed sample from the average value.

## Data Availability

Not applicable.

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
