# Peer review of "Concrete Examination of 100-Year-Old Bridge Structure above the K?odnica River Flowing through the Agglomeration of Upper Silesia in Gliwice: A Case Study"

_materials, 2021, doi:10.3390/ma14040981_

Round 1
Reviewer 1 Report
Ms. Ref. No.: materials-1098797 -peer-review-v1
Impact of the Chemical Composition of Water from KÅ‚odnica River flowing through the Agglomeration of Upper Silesia 3 on the 100 years old Bridge Structure in Gliwice
Reviewer comments:
SUMMARY
The manuscript deals with a good investigation on the effect of the Water Chemical Composition from KÅ‚odnica River on a 100 years old Bridge. Concrete were studied by means of pH, chloride content, salinity, XRD, 16 SEM examinations and metals determination using ICP-MS, HPLC-ICP-MS, CV-AAS. This is a topic that has not been widely covered in the literature, therefore, this a subject of great interest, but it is somehow limited in the analysis and application of these results.
MAIN IMPRESSIONS
This paper has an undeniable practical usefulness. However, from a scientific point of view, the following issues must be addressed: i) Research was performed in five concrete samples, then the sampling criteria should be defined (In theses studies an statistical analyses is normally given); ii) All the testing methods must be referenced in the paper, for instance, in line 499, The European standard EN 206:2013+A1:2016/prA2:2020 “Concrete - Specification, performance, production and conformity”, iii) Final remarks or conclusions should highlight the novelty of this paper. Could you please highlight the knowledge that the authors have contributed to the field? A proper conclusion should tell the reader what he could do with the newly acquired knowledge.
MORE DETAILED COMMENTS
Line 12: Could you please check the English grammar? For instance, “… analyzes the composition concrete, …” should be “… analyzes the composition of the concrete, …”
Line 21: Could you please check the English grammar? For instance, “… its soluble in acid part … “ could be “… soluble in dilute hydrochloric acid… “ or “… acid soluble components/portion …”.
Line 25: The abstract is written for the potentially interested reader. Then, you should give the main findings or conclusion.
Line 36: “has been previously briefly described (references below), .. “. : Could you please add the refences here?
Line 48: What about the steel reinforcement corrosion?
Line 57: Recent years? Ref. [7]: 20 years ago.
Line 59: In line 495 is said “ … bridge structures are heavily exposed to salinity due to the use of de-icing salts, …”. Therefore, could you please discuss similar conditions?
Line 81-87 & Fig.1: This paper does not deal with LCA. Therefore, you should consider removing this part or include a LCA of the bridge.
Line 90: Detailed study with five samples? Could you please justify this? How cubic meters of concrete has been used in the concrete bridge? Sampling criteria should be included.
Line 138: Add “1.2”. Line 177: Add “1.3”. Line 181: Add “1.3.1”. And so on.
Tables 1, 2, etc.: Replace commas by dots.
Table 2: sulphur or sulphates?
Table 2: Cl-
Line 269: Please, cite the reference and add the reference at the end.
Line 306: How cubic meters of concrete has been used in the concrete bridge? Five samples are enough to study the bridge performance? Sampling criteria should be included. Could you please justify the sampling criteria?
Line 306: Two water samples are enough to discuss 100 years of service life?
Fig. 10: What about the steel reinforcement corrosion? Could you please provide corrosion potential and / or corrosion rate measurements? How do the chloride content and / or carbonation influence reinforcement steel corrosion?
Line 452: pH about 12.8. It depends on several factors.
Line 503: The European standard EN 206:2013+A1:2016/prA2:2020 “Concrete - Specification, performance, production and conformity”, is one standard. Then, it should be in singular.
Line 503: If the European standard EN 206:2013+A1:2016/prA2:2020 does not apply, you should delete 499-507 and add the Polish regulation 100 years ago.
Line 503: “ ,. “. Delete “ . “.
Lines 495-496 and lines 508-509 say the same.
Table 6: Soluble chlorides in HCl soluble phases, could they contain chlorides from the HCl? Normally, acid soluble phases are determined by using HNO3? Could you please explain why have you used HCl? Could you please add the utilized standard?
Figure 15: “Theta” is missing.
Table 9: Could you please add subscripts here necessary.
Fig 16b: This is black. It is not possible to distinguish anything.
Lines 698-763: “4. Discussion” is 5% of this paper. No references at all. Could you please compare your results with the findings published by other authors?
Lines 764-774: Final remarks or conclusions should highlight the novelty of this paper. It is well-known that “ .. concrete did not absorb all types of metals …”. Please, check this in the literature and discuss it. You should review this point in the discussion giving references. A proper conclusion should tell the reader what he could do with the newly acquired knowledge. Answer the question "So what?".
References: Some of them do not have DOI.
RECOMMENDATION
In conclusion, Major changes have been proposed.
Reviewer 2 Report
Impact of the Chemical Composition of Water from KÅ‚odnica 2 River flowing through the Agglomeration of Upper Silesia 3 on the 100 years old Bridge Structure in Gliwice
The paper reports a study on the condition of a recently demolished bridge in Gilwice exposed to water from the Klondica river. The reporting of the condition of structures is a very useful tool to understand the influence of exposure conditions for both existing structures and for the design of new structures. The paper reports some very interesting results. Additional information on the test methodology and the changes in the environment with time is required but the paper is generally well written, though the English and grammar do require some minor editing.
Specific comments are
Line 36, can remove comment references below
Line 41 Give clear summary of the testing that was undertaken. Was analysed does not make details of what was actually done in terms of assessing historical data and the physical testing on the structure.
Line 54 pierce is this piers?
Line 69 does water refer to river water or rain water ? differentiate from the term groundwater to make clear
Line 170 The sentence Now leave does not relate to the previous information, please check the meaning of this sentence in context of the paper
Line 200 – when stating the water is clearer does this simply mean visibly clearer, ie less silt/soilds or does this relate to other pollution?
Line 228 some of the text appears to be missing after the …and
Line 239, please explain why re-loading lead to the river being high in slime, the current explanation is not clear.
Line 306 the first paragraph in the materials section can be removed as the information is given previously in the introduction to the rationale for the operation and closure of the bridge.
Line 312 – what are the dimensions of the concrete samples, ie are these cores, dust drillings, depth etc
Similarly give details of volume of water samples
Line 344 To what size was the concrete powder crushed, ie passing 75 micron sieve?
Line 352, please give details of any standards used in the chemical analysis or references to methodology adopted from previous papers, ie moisture content, pH
Line 351, what is a mine-thrower, please give equipment details as in name/make
Line 378, what is energy used for SEM analysis
Line 460 – what is the relationship between the leaching of the Ca and carbonation on the pH. At the top of the bridge this is likely to ne more exposed to carbonation from traffic and thus the lower pH may be a function of both the leaching and the carbonation. This should be addressed at some point in the paper, either in the results or the later discussion. At present each individual set of data is discussed but the combination of different mechanisms such as chloride ingress and carbonation etc are not commented upon as the condition of the structure is likely to be due to the interaction of a number of deterioration mechanisms present from the environment.
Line 527 what is the other method for determining pH?
Line 578 repeat specimens in the sentence
Line 580 what is the evidence that calcite is from corrosion products, ie carbonation not the aggregate
Line 684, when commenting on the increases and decreases in the metal content in the river (and the chloride content) is there evidence or information about the historical content to show how much these have reduced or increased over time.
Line 719 What is the sulphate content in the river water/ground water. Sulphate content would be expected to be low from de-icing salts and only the chloride content is given for the river water. The microscopy data shows a limited ettringite content thus there appears little evidence for high sulphate exposure from the bridge and the cracking of concrete due ettringite formation and sulphate attack. Sewage outlets are mentioned, is there evidence for these leading to high sulphate exposure?
Line 756, the full stop after is appears to be in wrong place
Line 759 please check meaning of this sentence.
Conclusions - The conclusion needs to be expanded to summaries the key findings of the study for example where does statement that CEMI was probably sued derived in the paper as no mention is made of Cem! Cement whereas all the following statements are very brief and need farther explanation, yet these is no mention of the martials that are seen to be absorbed by the cement, nor the low pH of the cement etc. The conclusions should be rewritten
Reviewer 3 Report
After a careful reading of the full manuscript I would conclude that while it obviously appears that a lot of work and effort have been put in this paper, unfortunately, there are many critical points that need to be improved:
I would suggest modifying the title of the work. In my opinion, I am not sure if the majority of the authors would be searching for the local areas.
From reading the abstract, it is not clear what is the novelty of this article or why such an investigation is needed. Why this specific bridge was chosen?
Generally, I would suggest improving the written style and correcting many grammar mistakes and typos.
The Introduction should be revised. References are missing in the first part. Additionally, I would prefer to start the introduction with explaining why this topic is important to be addressed, what are the „many physical and chemical environmental factors“ etc. Usually, what the presented manuscript deals with it’s explained at the end of the introduction part. Here, the organization of the introduction is confusing to the reader.
„The thaumasite corrosion of concrete bridge structures has been widely described in recent years.“ It should be supported with several references.
„However these bridges were not built above rivers, therefore the mechanism for concrete corrosion is not similar to that of the KÅ‚odnica river due to the absence of chloride and sulfate ions.“ References?
In my opinion, more related studies should be cited and compared with the studied issue.
„We are not sure of the effectiveness of the adopted maintenance strategies and repair or rehabilitation methods, for both the individual components as well as the entire bridge.“ I am not sure if I fully understand the meaning of this sentence.
The section Bridge destruction, processes and the life cycle of the bridge is too general and doesn’t provide any new insights into studied problem. Some related studies by other researchers and their results should be discussed here.
In my opinion, Figures 1 and 2 are very general and not necessary.
„The concrete specimens were collected in fall 2019 and the water from the river in spring 2020.“ Why the time difference?
There are many controversial and not fully accurate sentences, which should be revised, such as “Many rivers in the world are heavily polluted by industry or by human daily living needs. This effect has been widely studied and accepted [35–48].” What is meant being accepted in this case? I would recommend a careful reading of the full introduction and modifying all vague statements.
Lines 164 – 167 should be supported by references.
Lines 171 – 174 should be supported by references.
The section about KÅ‚odnica River should be evenly supported by references. It is not appropriate to merge all references only at the end of the section. Moreover, the text is too general.
“Dense proximity of these factories has a great impact on the environment” What is an approximate distance between the factories? References?
“The bottom of the KÅ‚odnica valley is flat and wet.” It doesn’t sound scientific.
“This is evidenced by high pollution rates in rivers flowing through this area, in both their sediments and in the waters of the lakes to which they flow.” What does it mean “high pollution rates”? + References
Figure 3 is not clear. It doesn’t show the 10 tributaries which referred to by this picture.
In general, the introduction part until the section about the examined bridge seems to be not ideally organized and doesn’t provide proper explanations about the studied topic, doesn’t highlight the main problems and it is way too general for the Reader.
Table 4: It is not clear which samples are related to which comments.
„The moisture of the concrete was tested immediately after heating in an oven at 105 °C to achieve a constant mass.“ It is not clear if the samples were powdered or crushed.
„Concrete composition and pH examination.“ The title contains concrete composition, while the measurement of the moisture content is described along with pH tests.
The description of the used program for XRD measurements is missing and should be added.
The length of the description of the used methods should be balanced. Why is the metals and pH determination explained more in details compared to the other methods? The choice of the used methods and their settings should be explained.
Table 5, how many measurements were done for each specimen? The standard deviation should be added.
For the pH results, it would be worthy to add the depths from which the samples were collected.
I am not sure of I understood well why there are two different measurements of pH (table 5 + table 8).
XRD results: It seems that quartz was the most dominant component. And thus, it is very difficult to analyze the composition of such samples only with this method. Why thermal analysis was not used? It would have been perfect if looking for the remaining portlandite.
Figure 15, if all graphs are shown with a certain shift, I’d use a general unit of intensity (a.u.) instead of counts.
„An accumulation of needle-like sulfate salts, possibly ettringite Ca6Al2(SO4)3(OH)12·26H2O or even thaumasite: Ca3Si(OH)6(CO3)(SO4)·12H2O phases, were observed in the pores, located here in a small amount, and not detected using diffractometry, however, they are seen in the SEM micrographs in Figure 18.“ Was it confirmed by XRD?
A general comment, the article contains many parts that are very repetitive.
The discussion doesn’t contain almost any references and it is not providing any new insight into the studied problem.
„The concrete described was probably made with CEM I.“ Based on which results?
„It seems necessary to examine the next bridges over the KÅ‚odnica River.“ How often?
„In the face of disasters, it is very important to expand research on bridges. Maintaining a numerical database of building objects is essential.“ Both sentences are very general and vague.
The number of self-citations is in a reasonable range: SÅ‚omka-SÅ‚upik 3; Podwórny 1; Grynkiewicz-Bylina 1; Salamak 2; Bartoszek 1
Reviewer 4 Report
The research is very much interesting. However, the authors discussed the results superficially without concluding. This is the reason why the manuscript is more of a report rather than a research paper. The authors should be able to screen the observations and test results and clearly define the reason for the deterioration of the bridge sections. Please remove unnecessary information and refine the manuscript as well.
As said in the abstract “The goal of this work was also drawing attention to the need for chemical testing of bridge elements.” However, after reading the paper, one can conclude that chemical testing of the bridge was useless.
Unfortunately, my comment is to reject the work in the current format. I strongly suggest the authors reformat the manuscript thoroughly and make a clear conclusion.
Some minor comments from the beginning of the paper are as below:
Line 35: “The issues surrounding the destruction of bridges by water has been previously briefly described (references below), however, the corrosion of concrete under the influence of the river water has not been dealt with.” This sentence is incorrect or at least inaccurate. Many papers studied the adverse effect of the river on the concrete deterioration.
Line 46: “the content of ions that corrode the concrete, in the river” I believe the corrosion is an electrochemical process for metals (please check the handbooks). Therefore, corrosion of concrete is an inaccurate term for the deterioration of concrete. However, corrosion of steel is a meaningful term. Please rectify this concern all over the text.
Line 61: “Exemplary studies on the recently collapsed Polcevera Bridge in Genoa (Italy) have demonstrated that the hostile environment caused the degradation of the bridge to develop much faster than expected.” What was the reason for the collapse of the bridge in Genoa? Wasn’t due to corrosion of pre-stressing cables?
I am eagerly waiting to get an elevated format of the work.
Reviewer 5 Report
The authors have done a meaningful study for investigating the influence of the chemicals in water on the concrete. The in situ works are vital to sciences as well as engineering aspect.
However, this study has some problems needing to be revised.
1. For introduction part, the background of corrosion of concrete, the hazard of absorbing heavy metals into concrete, and chloride ion etc. shall be interpreted. Alternatively, the references must be cited into the introduction part for the evidence confirmation. 2. Line 41, which one “was analyzed”? It should be formal. 3. Line 42-48, the start of the sentence, the letter should be capital and at the end of the sentence the dot shall be shown. 4. In short, the purpose of this study was seem to be like from the authors’ unidirectional perspective. Thus, the introduction must be revised 5. For Materials and Methods part, line 306, “the research material” why this word in singular? 6. Line 307, “The Demolished bridge which used to traverse it” should be re-considered. 7. Line 330, the authors need to show the reason why two location points in the river i.e. W1 and W2 were investigated? 8. Line 334, the powder should be shown the production and storage process. 9. Why were the pH determination tests performed many times in this study? The reason should discuss before the presence of the pH tests. 10. For all experimental test method, the information of machines, tools, technical parameters, and operation must be shown. 11. Line 357, Free chloride content examination must be shown at the subject. 12. Line 358-366, it seems to be like this method is generated by authors due to without any reference although ASTM C1218 is known as the standard of determining the free chloride amount in mortar/concrete. Thus, the authors’ method may be not reliable because test conditions are capable of affecting water soluble chloride determination, which are ascribed to: a. The water and concrete powder mix should be stood for 24 hours. b. Nitric acid (1:1) should be employed. c. The free chloride content data in this study may not correspond to other studies. 13. Line 384, this sentence should be revised due to misspellings. 14. Line 390-391, “This method was chosen because … many elements” should be confirmed by other references. 15. For Metals and pH determination part, the types of metals identified method should be presented individually and titled. 16. Line 421-422, “the samples” for which experiment? 17. Line 427, “a pressure filtration device and an appropriate amount … the elements” the number of optimized amount should be discussed particularly. 18. In short, the layout of Part 2 needs to be re-written clearly and coherently for easier understanding for readers. 19. Line 446-450, if a part of sand having the less amount of amorphous morphology be able to react with cement and form the CSH as a hydrated product, can the definition of “soluble phase” use? 20. Line 452, what is the meaning of “ca.”? 21. Line 487, what is the cause of black aggregate to the railings? 22. In Figure 12, why is the dark color shown in bucker 3? 23. Line 499, why don’t authors make the reference for this citation? 24. Line 505, with “the” consideration should be edited. 25. Line 513-514, why is the large number of chloride ions present in sample 2 and the lowest content of chlorides is found in the sample 5? The reason must be discussed clearly. 26. For table 6, what is difference between “chloride content in concrete” and “chloride content in soluble phases”? Soluble phase is also in concrete, isn’t it? 27. Figure 13, chloride content including binding chloride, free chloride, and total chloride. Which type of chloride the authors want to show? 28. Line 531, why is the healthies concrete at location 3 and 4? It is critical to discuss this point clearly. 29. Line 560, “sufficient” has already included “so” inside. Thus, the writing style should be revised. 30. For Table 9, it should be changed the format of marking the presence of chemicals over samples to bullets i.e. x, o, √, etc. instead of using colors for easier observe in white-black reading. 31. Why was only Cu detected in specimen 4? 32. Line 622, what is the element “Zu”? the author must show Zu in which data part. 33. What amount and type of metals/heavy metals cause the deterioration of concrete? The degree of the hazard of aggressive metals should be shown. 34. Line 664, why do the authors know that “lead and arsenic were more difficult to absorb”? 35. Line 685-687, “Nickel penetrated the ground … foundations of the bridge (specimen 5)” is an affirmation. Thus, it need to add some references for proving this point. 36. Discussion part is for explaining to the Results part. Thus, the content from line 699-713 should be in Introduction part. In addition, the discussion part must explain and justify the data in the results part instead of showing the data with less justification for itself. 37. For conclusion, the outlook of this part should be in one paragraph, which induces the layout of the conclusionlosing balance. 38. Spelling mistakes and style should be revised.Author Response
Please see the attachment

Round 2
Reviewer 1 Report
Line 416: EN ISO 12570:2000/A2:2018 Hygrothermal performance of building materials and products - Determination of moisture content by drying at elevated temperature (ISO 12570:2000/Amd 2:2018). https://standards.iteh.ai/catalog/standards/cen/0a862afc-7d17-4bee-8592-d232d001e573/en-iso-12570-2000-a2-2018
Line 600: Editorial: 86-91].
Line 600: Old comment: Line 503: The European standard EN 206:2013+A1:2016/prA2:2020 “Concrete - Specification, performance, production and conformity”, is one standard. Then, it should be in singular.
Line 815: “4. General Discussion” & Line 544 “3. Results and Discussion”: Discussion twice?
Line 946: Why have you removed the chapter “5. Conclusions”? The conclusion should conclude the paper and is written for the reader who already has read the paper. In other words: most readers have read the paper when they read the conclusion.
Author Response
Thank you very much for revisions. Best wishes and stay safe!
Authors.

Reviewer 3 Report
- The current title of the manuscript seems more accurate and the abstract has been improved.
- Generally, I would suggest improving the written style and correcting many grammar mistakes and typos. This comment is still valid.
- In my opinion, more related studies should be cited and compared with the studied issue. Answer: “Of course, you are right. To be honest, there aren't many similar cases reported in the literature.” One of the reasons, why this information should be included in the text. I’d recommend using the phrase: “To our best knowledge, there is no literature concerning…”
- The section Bridge destruction, processes and the life cycle of the bridge is too general and doesn’t provide any new insights into studied problem. Some related studies by other researchers and their results should be discussed here. Answer: “The philosophy of this article has changed. Now the durability of the bridge is being described rather than the influence of the river components. And the bridge specialist (co-author) says it is very important.” The bridge specialist thinks that changing the strategy of this section is important or the two mentioned Figures are important to be included? It is not clear. Honestly, the Figure 1 brings no additional information than what is already in the text. The same for Figure 2. For the Figure 2: There is only one related sentence justifying the presence of the figure.
- There are many controversial and not fully accurate sentences, which should be revised, such as “Many rivers in the world are heavily polluted by industry or by human daily living needs. This effect has been widely studied and accepted [35–48].” What is meant being accepted in this case? I would recommend a careful reading of the full introduction and modifying all vague statements. Answer: "Indeed, this sentence is very strange. I don't know what's going on anymore. Carefully read the introduction is required. Without a doubt. “This effect has been widely studied and accepted” –was deleted.” The authors agreed that the careful reading of the introduction (= leading to modifying the vague statements) is required without a doubt. But I am not sure if we understood each other here since not much has changed. This previous comment is still valid.
- The section about KÅ‚odnica River should be evenly supported by references. It is not appropriate to merge all references only at the end of the section. Moreover, the text is too general. Answer: “Yes, because the river itself is not a research area. The information at this point (1.3.1) is a conglomerate of articles mainly researching the composition of its water. Literature references would appear by each detail, which would look bad.” Supporting the used information taken from the other authors would look bad in a research paper? Really?
- “This is evidenced by high pollution rates in rivers flowing through this area, in both their sediments and in the waters of the lakes to which they flow.” What does it mean “high pollution rates”? + References. Answer: “Instead of the word “rates”, “indicators” was introduced”. Reference is still missing.
- Figure 3 is not clear. It doesn’t show the 10 tributaries which referred to by this picture. Answer: “The sentence was changed, added: „it has more than 12 tributaries” I am not sure if we understood each other here. My concern was about the fact that if Figure 3 is supposed to be demonstrative to such a sentence, these 12 tributaries should be highlighted in the pictures. If not, I believe that the Reader can find a map of Poland by herself/himself, and thus, this map is not necessary to be shown in the manuscript.
- In general, the introduction part until the section about the examined bridge seems to be not ideally organized and doesn’t provide proper explanations about the studied topic, doesn’t highlight the main problems and it is way too general for the Reader. Still valid comment.
- „The moisture of the concrete was tested immediately after heating in an oven at 105 °C to achieve a constant mass.“ It is not clear if the samples were powdered or crushed. Answer: “Liquid samples of the water from the KÅ‚odnica river (W1, W2) were ...immediately tested.” The answer is more confusing that the previous statement.
- Table 5, how many measurements were done for each specimen? The standard deviation should be added. Answer: “These types of tests are performed on averaged single samples, so one” In my opinion, one sample/one measurement is not enough for a research article. It should be stated honestly in the manuscript.
- For the pH results, it would be worthy to add the depths from which the samples were collected. Answer: “Yes, but the thickness of material (specimens 1-4) was of about 5 cm. Bridge plate was thin and moisture of the concrete depends on the weather... I have a problem with it too, because I usually do laboratory tests (controlled)” In this case I don’t understand why this information is not included in the manuscript.
- XRD results: It seems that quartz was the most dominant component. And thus, it is very difficult to analyze the composition of such samples only with this method. Why thermal analysis was not used? It would have been perfect if looking for the remaining portlandite. Answer: “ Podwórny is a a very good specialist in analyzing the phase composition. Thermal analysis would not reveal so many trace minerals. Portlandite at such a low pH (less than 12.5) was notexpected-the calcium carbonate formed from it. I was rather expecting thaumasite, but no sulfates were found.” Did prof. Podwórny try to quantify the found phases? I believe, that it will be an interesting information to the Reader.
- Figure 15: It seems that samples numbered 0, 3 and 5 were measured using a different program, as the area between 5 and 10 2theta is different compared to other samples. Could the authors explain why?
- A general comment, the article contains many parts that are very repetitive. Answer: “Of course, the article will be carefully read again. We'll remove the recurring issues.” Which parts have been modified in accordance to this authors’ answer?
- The discussion has been improved.
- Conclusions: In my opinion this part should be longer and summarize the main findings.
Author Response
Please see the attachment.
Thank you very much for the revisions. Stay safe!
Authors.

Reviewer 4 Report
Thanks to the authors for revision of the manuscript.
Author Response
Thank You very much! Best wishes and stay safe!
Authors.
Reviewer 5 Report
authors have responded to my all concerns, therefore, now it can be accepted for publication.
Author Response

(The authors gave the same response as above.)
